

# How well do the regional atmospheric and oceanic models describe the Antarctic sea ice albedo?

Kristiina Verro[1,9], Cecilia Äijälä[2], Roberta Pirazzini[3], Ruzica Dadic[7,8], Damien Maure[4,5], Willem Jan van de Berg[1], Giacomo Traversa[6], Christiaan T. van Dalum[1], Petteri Uotila[2], Xavier Fettweis[4], Biagio Di Mauro[6], and Milla Johansson[3]

[1]Institute for Marine and Atmospheric Research (IMAU), Utrecht University, Utrecht, the Netherlands
[2]University of Helsinki, Institute for Atmospheric and Earth System Research / Physics, Helsinki, Finland
[3]Finnish Meteorological Institute, Helsinki, Finland
[4]SPHERES research unit, Geography, University of Liège, Liège, Belgium
[5]Univ. Grenoble Alpes, CNRS, IRD, Grenoble INP, IGE, 38000 Grenoble, France
[6]Institute of Polar Sciences, National Research Council, Milan, Italy
[7]WSL Institute for Snow and Avalanche Research SLF, Davos, Switzerland
[8]Victoria University of Wellington, Antarctic Research Centre, Wellington, New Zealand
[9]National Centre for Climate Research, Danish Meteorological Institute, Copenhagen, Denmark

**Correspondence:** Kristiina Verro (kve@dmi.dk)

**Abstract.** A realistic representation of the Antarctic sea ice surface albedo, especially during the spring and summer periods, is essential to obtain reliable atmospheric and oceanic model predictions. We used regional climate (HCLIM, MAR, RACMO), regional oceanic (MetROMS-UHel, NEMO) models and ERA5 reanalysis to investigate how well these models describe the basic sea ice characteristics: sea ice albedo, snow and ice thickness. We analyse models against a range of observations, including in-situ measurements from the ISPOL (Weddell Sea, Dec. 2004) and Marsden (McMurdo Sound, Nov. 2022) field campaigns, as well as drone and satellite data. Models perform well in reproducing the sea ice in certain conditions: during the ISPOL campaign, characterised by thicker snow cover and mild weather that resulted in daytime melt-driven albedo changes and nighttime refreezing in the snow-covered sea ice most models did well; MetROMS-UHel, NEMO, HCLIM and MAR reproduce mean values found in observations, and MetROMS-UHel captures even the observed diurnal albedo variability. However, all models had difficulty reproducing the sea ice conditions in the McMurdo Sound. The observed mean surface albedo was largely influenced by variations in drifting snow accumulation patterns over very thin (few to few tens of cm) snow cover and most models clearly overestimated the albedo. Over the colder and drier sea-ice regions with thin or patchy snow cover, the key issues affecting the accuracy of albedo models are the treatment of fractional snow cover and the snow albedo dependence on snow depth. Over the broader Weddell and Ross seas, sea ice albedo is primarily determined by sea ice concentration fields. HCLIM, MAR, and RACMO rely on ERA5 input for sea ice concentration fields, whereas MetROMS-UHel and NEMO calculate them internally, resulting in differences in both sea ice concentration and albedo patterns. Albedo parameterisations are still relevant: RACMO and ERA5 predict significantly darker sea ice over the Weddell Sea during the ISPOL campaign, while their predictions align better with observations over the Ross Sea during the Marsden campaign. Sea ice albedo is typically parameterised in models as a function of one or more variables, including air temperature, surface



temperature, snow/ice type, snow grain size, snow depth, density, sea ice thickness, cloud cover fraction and solar zenith
angle. The simplest approaches, like those in ERA5 and RACMO, rely on prescribed sea ice albedo values based on Ebert
and Curry (1993). In HCLIM, the intermediate-complexity snow model determines snow reflectivity based on snow grain size
distribution, which is only a function of snow density, and the bare ice albedo follows a simple temperature-based relationship.
When more sophisticated radiative transfer schemes are applied, albedo is calculated based on the inherent optical properties

of the surface, such as in MetROMS-UHel. Integrating advanced radiative transfer models to the regional climate or ocean
models, represents a significant advancement in simulating surface processes.

## 1  Introduction

The Antarctic sea ice zone covers approximately $19 \times 10^6$ km$^2$ in winter and $3.5 \times 10^6$ km$^2$ in summer. Sea ice plays a crucial
role in the polar climate due to its high albedo compared to the open ocean, significantly influencing the surface energy budget

and, consequently, the mass and heat balance of the sea ice. This can initiate a positive feedback loop, where the loss of sea
ice and snow cover reduces albedo, further increasing surface temperatures, accelerating the melt and increasing sea ice loss.
This positive surface albedo feedback has been shown to contribute to Arctic amplification in both models and observations
(Previdi et al., 2021, and references therein), but is less reported in the Antarctic (Casado et al., 2023).

Accurately representing the properties of sea ice, such as the albedo, ice and snow thickness, and their interactions with the

atmosphere and ocean is crucial for reliable predictions in polar regions. These properties of sea ice can vary significantly. Sea
ice consists of ice floes of varying thickness, typically covered by snow, and is often fragmented by cracks, leads, and polynyas.
The surface albedo of sea ice depends on snow and ice characteristics and can range widely – from 0.9 for dry fresh snow, to
0.4 for melting bare ice, but the spatial average albedo is highly affected by the fraction of open water, as it has an albedo of
0.06.(Warren, 1982; Perovich and Gow, 1996; Pirazzini, 2004; Light et al., 2022). The albedo depends also on planetary and

atmospheric factors, such as the solar zenith angle and cloud cover.

Simple parameterisation schemes for sea ice are traditionally employed in regional climate model applications. This is the
case for the models used in this study: HCLIM (Belušić et al., 2020), MAR (Gallée and Schayes, 1994), RACMO (van Dalum
et al., 2024). In these models, sea ice concentration is prescribed, while sea ice thickness is either fixed (as in RACMO) or
thermodynamically evolving (HCLIM, MAR), while neglecting the dynamical processes. In ocean modelling, more advanced

dynamic-thermodynamic sea ice models are often applied. Examples of these include NEMO (Nucleus for European Modelling
of the Ocean, Madec et al., 2023) with the sea ice engine SI$^3$ (Sea Ice modelling Integrated Initiative, Vancoppenolle et al.,
2023), and ROMS (Regional Ocean Modelling System, Shchepetkin and McWilliams, 2005) coupled with the sea ice model
CICE (Community Ice CodE, Hunke et al., 2022) by (Debernard et al., 2017; Naughten et al., 2017; Äijälä et al., 2024).

Both the regional climate models and the regional ocean models rely on sea ice albedo parametrisation recommendations,

such as those given in Ebert and Curry (1993); Perovich and Gow (1996); Curry et al. (2001); Pirazzini (2004); Liu et al.
(2007); Briegleb et al. (2007). Sea ice albedo is typically parameterised as a function of one or more variables, including air
temperature, surface temperature, snow/ice type, snow grain size, snow depth/density, sea ice thickness, cloud cover fraction





and solar zenith angle. The complexity of these parameterisations varies depending on the intended application of the model. In the simplest cases, prescribed average sea ice albedo values are used, as in ERA5 Hersbach et al. (2020) and RACMO. When more sophisticated radiative transfer schemes are applied, albedo is calculated based on the inherent optical properties of the surface (e.g. snow model CROCUS used in MAR, Brun et al., 1992).

Curry et al. (2001) and Liu et al. (2007) pointed out, that most climate modellers have justified using simple snow/ice albedo parameterisations because of the lack of observations against which to evaluate them. Indeed, detailed sea ice albedo-specific studies, which compare both observations and models, are rare. Recently, Jäkel et al. (2023) tested the coupled Arctic atmosphere-ocean-sea ice model HIRHAM–NAOSIM model in the Arctic against MOSAiC campaign in-situ observations, aircraft measurements, and satellite images. They found large (>0.1) discrepancies between albedo observations and models during the melt season and concluded that the surface-type parametrisation contributes significantly to the bias in albedo. Furthermore, the ERA5 sea ice albedo, or more exactly, the sea ice albedo of the Integrated Forecasting System (IFS) model of the European Centre for Medium-Range Weather Forecasts (ECMWF), has been identified as a limitation by several studies over the Arctic region (Pohl et al., 2020; Müller et al., 2024; Batrak et al., 2024). Batrak et al. (2024) investigated the albedo of sea ice in ERA5 and the Copernicus Arctic Regional Reanalysis (CARRA) dataset and found spring-time albedo errors around 0.06 and 0.14 respectively, computed against the CLARA-A2 satellite retrieval product (Karlsson et al., 2017). Therefore, both reanalysis datasets have issues over the Arctic sea ice, but they determine sea ice albedo differently: CARRA uses modelled albedos from the HARMONIE-AROME NWP system (Bengtsson et al., 2017), while ERA5 relies on monthly albedo values from Ebert and Curry (1993), which are valid on the 15th of each month and linearly interpolated for the other days (ECMWF, 2016).

Studies over Antarctica are crucial due to the distinct differences in sea ice characteristics between the polar regions. In the Arctic, summer snow cover melts first, leading to the exposure and subsequent melting of bare ice from above. In contrast, Antarctic conditions are characterised by colder, drier air and relatively greater heat flux from the ocean, causing sea ice to melt from the bottom up while retaining its snow cover for a longer period (Brandt et al., 2005b).

This paper evaluates a set of regional climate models (HCLIM, MAR, RACMO), regional oceanic models (MetROMS-UHel, NEMO), and the fifth generation ECMWF reanalysis for the global climate and weather (ERA5 Hersbach et al., 2020) to assess how accurately they represent fundamental sea ice characteristics: sea ice albedo, snow thickness, and ice thickness. Each of these models is different in how they handle sea ice and surface albedo. We aim to bridge the models to observations, by comparing the model outputs to sea ice observations of various spatial scales, from in-situ measurements from field campaigns to drone and satellite observations. We focus on the Weddell Sea and the McMurdo Sound/Ross Sea domains, which exhibit distinctly different sea ice conditions. Our study examines the spring and summer seasons, when solar insolation is high, and surface albedo plays a critical role in controlling the surface energy budget. The model validation was conducted using albedo, snow, and sea-ice observations over the Antarctic sea ice. Surface-based and drone-based measurements collected during the Ice Station Polarstern (ISPOL, Dec. 2004, Weddell Sea,  Hellmer et al., 2008) and the New Zealand Marsden field campaign (Nov. 2022, McMurdo Sound,  Dadic et al., 2023) were used. To bridge between the limited footprint of the in-situ measurements and the larger footprint of the models, satellite observations were also used.





The paper is structured as follows. Section 2 describes the in-situ and satellite observation data used in this study. Section 3 provides an overview of the models used. The results are presented in Sect. 4, covering atmospheric conditions (Sec. 4.1), albedo time series (Sect. 4.2), snow and ice thicknesses (Sect. 4.3), sub-grid sea ice characteristics (Sect. 4.4) and comparisons to satellite albedo products (Sect. 4.5, 4.6). Section 5 is the discussion, where the results are explained, connected to other research, and their importance and limitations are explored. Section 6 summarises the findings and presents key takeaways of the study.

## 2 Observational campaigns and satellite images

Data from surface-based and drone-based measurements collected during the ISPOL and Marsden field campaigns were used for model validation. Figure 1 highlights the locations of these campaigns. To complement the localised in-situ measurements and align them with the larger spatial coverage of the models, Landsat 9, Sentinel 2 and CLARA-A2 satellite observations were integrated into the analysis.

### 2.1 The ISPOL campaign

The ISPOL campaign took place in the Weddell Sea from 27 Nov. 2004 until 2 Jan. 2005. The main aim of the campaign was to gather data on physical and biogeochemical interactions between the atmosphere, ice and ocean during the transition from austral winter to summer. The ISPOL experiment was mainly conducted on a single ice floe in the Weddell Sea, where the research vessel (RV) Polarstern was moored. The ice floe was located around 68°S, 55°W, drifting 290 km during the experiment with a 98 km south-north displacement (Fig. 1, Hellmer et al., 2006; Hellmer et al., 2008; Vihma et al., 2009). The entire ISPOL floe comprised two-meter-thick second-year ice (SYI) covered by 0.8 meters of snow, along with locally formed and advected first-year ice (FYI) featuring modal thicknesses of 0.9 and 1.8 meters, respectively, and topped with 0.3 meters of snow. (Hellmer et al., 2006). Broadband albedo was measured from pyranometers installed at about 1.5 m above the surface. The AWI measurement station was operated over the FYI, and the FIMR station obtained readings from the SYI. After the FIMR radiation station was relocated on 17.12 and 27.12 due to the ice flow break-up, thick snow sites were chosen to maintain consistency in the radiation time series. Therefore, we refer to the measurements from the AWI and FIMR radiation sites as FYI and SYI, respectively.

The surface albedo was calculated as the ratio of the reflected and incoming shortwave radiation. The accuracy is approximately 3% for the shortwave radiation measurements (Vihma et al., 2009). The measured ice floe was surrounded by leads and fragmented ice. The weather was warm, with air temperature mostly above -5°C and even around zero degrees during the first week of December. The meteorological observations of Bareiss and Görgen (2008), and observed properties of snow and sea ice from Vihma et al. (2009); Haas et al. (2008); Nicolaus et al. (2009) are used to assess the performance of the regional atmospheric and oceanic models described in Sect. 3.



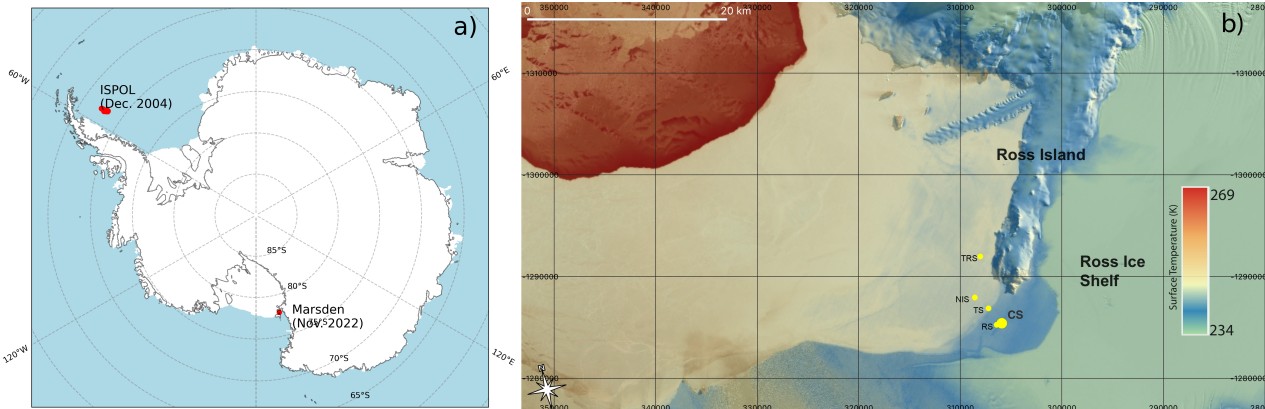

**Figure 1.** a) The locations of the ISPOL and Marsden campaigns. b) Locations of the camp site (CS), Ridge Site (RS), Transition Site (TS), New Ice Site (NIS), and Turtle Rock Site (TRS) in McMurdo Sound during the Marsden field campaign. The image shows an overlay of Landsat surface temperatures over a Landsat grayscale visible image on October 10, 2022. The temperature scale is approximate only because the overlay is slightly transparent, which alters the colour.

## 2.2 The Marsden field campaign

A field campaign on the land-fast sea ice in the McMurdo Sound was carried out in November 2022 to assess the physical
properties and spatial variability of snow on the land-fast sea ice. The campaign was part of the New Zealand Marsden Fund project "Can Snow Change the Fate of Antarctic Sea Ice?" (Dadic et al. (2023), Dadic et al. in prep). Figure 1 shows the locations of the camp site (CS), Ridge Site (RS), Transition Site (TS), New Ice Site (NIS), and Turtle Rock Site (TRS) of the Marsden field campaign in McMurdo Sound, Antarctica.

The albedo observations used in this study were collected over two areas of different sea-ice thickness (∼2.3 m CS and
1.2 m NIS) located at about 2 km distance from each other. The thickest sea ice formed earlier in the winter and was closer to the edge of the Ross Ice Shelf. Broadband albedo was measured on both locations using a pair of Kipp and Zonen CM4 pyranometers installed on a drone, while continuous time series of broadband albedo was measured with a pair of Kipp and Zonen CMP22 pyranometers installed at about 1.2 m above the surface in a fixed station at the CS over the thicker ice. The drone's pyranometers are embedded into gimbals that ensure their horizontal alignment. However, continuous adjustment of the
alignment generates vibrations that, when summed to the instrumental error, cause a total uncertainty in the measured albedo of the order of +-3%. The uncertainty of albedo measured with the fixed CMP22 pyranometers is around +-1%. Snow depth was measured in snow pits and along horizontal transects using a magnetostrictive device (magnaprobe), while ice thickness was measured in ice holes and along transects with an electromagnetic induction device (EM-31) carried on a sledge. The campaign was characterised by a steady temperature increase from -25°C to -3°C with a snowstorm in the last week of the
campaign (27th and 28th of November).





## 2.3  Satellite products

Data from two multispectral satellites, Landsat 9 and Sentinel 2, are used to estimate the high-resolution broadband surface albedo over the Marsden field campaign site. The instruments onboard Landsat 9 and Sentinel 2 provide images at different spatial resolutions, depending on the wavelength of the spectral band, but the broadband albedo products can be calculated

at 30 m and 20 m resolutions, respectively. The images were acquired in different channels in visible–short-wave infrared wavelengths (452-2294 nm and 458-2280 nm respectively) during the period of the Marsden expedition. To obtain the broadband albedo values, the Liang (2001) algorithm of narrowband to broadband albedo conversion was applied, which uses the blue, red, near-infrared and short-wave infrared bands to estimate the broadband albedo. For the Sentinel 2 image, the L2A reflectance product with already calibrated bands is used. However, the Landsat 9 L1 image went through further processing,

using the methodology proposed by Traversa et al. (2021) for high latitudes. This involved several correction steps (radiometric calibration and zenith angle, atmospheric and topographic corrections) to estimate the surface albedo. Correction for the anisotropy was not applied, since existing correction models are not able to suitably account for albedo variations at high solar zenith angles Traversa and Fugazza (2021). Furthermore, the CLARA-A3 blue-sky albedo product (Karlsson et al., 2023), which has a coarser resolution of 25 km, is used to evaluate the albedo variability over the whole Weddell and Ross Seas. Two

high-resolution satellite images from the Marsden campaign site and two lower-resolution, broader-scale satellite images –one from the ISPOL site and one from the Marsden campaign site – are utilised. The commonly used albedo products from the Moderate Resolution Imaging Spectroradiometer (MODIS, e.g. Hall and Riggs, 2021) do not cover the open ocean, thus we are not able to use them.

## 3  Regional oceanic and atmospheric models

The observations are used to assess the albedo modelled by MetROMS-UHel and NEMO ocean models, the regional atmospheric models HCLIM, MAR and RACMO, and the ERA5 reanalysis product. The sea ice and snow-related characteristics of these models are summarised in Table 1.

### 3.1  MetROMS-UHel

MetROMS-UHel (Naughten et al., 2018; Äijälä et al., 2024) is a coupled ocean-sea ice model. It consists of the terrain fol-

lowing the regional ocean model ROMS (Regional Ocean Modelling System, Shchepetkin and McWilliams, 2005) dynamic-thermodynamic sea ice model CICE (Hunke et al., 2022). The model uses the development version 3.7 of ROMS code and version 6.3.1 of CICE for sea ice. The model is run on a 1/4° grid with the northern border at 30° S and a relocated south pole. This gives a Cartesian resolution of 8-12 km on the continental shelves around Antarctica. ERA5 reanalysis is used for atmospheric forcing. A detailed description of the model setup can be found in Äijälä et al. (in prep.).

The sea ice has 5 thickness categories, 7 vertical ice layers, each having one snow layer. The lower bounds for the ice thickness categories are 0.0 m, 0.64 m, 1.39 m, 2.47 m, and 4.57 m. The model uses elastic-viscous-plastic formulation for





| Model | Horizontal Resol. | sea ice and snow | ice thickness categories (lower, m) | optical properties dependence |
|---|---|---|---|---|
| MetROMS-UHel | 1/4°, northern border at 30° (10–12 km at ISPOL) | CICE: dynamic (elastic-viscous-plastic); thermodynamic (mushy layer, melt ponds, refreezing); 7 ice, 1 snow layer | 0.0, 0.64, 1.39, 2.47, 4.57 | ice thickness and ice absorption/scattering; snow depth and snow optical grain size; melt pond depth and area fraction; solar zenith angle |
| NEMO | 1/4°, the northern border at 55° (10 km at ISPOL) | SI3: dynamic (viscous-plastic); thermodynamic (mushy layer, melt ponds, refreezing) | 0.0, 0.5, 1.1, 2.1, 3.7 | ice thickness, snow depth, cloud fraction, surface temperature; overcast sky albedo asymptotic value of 0.85 (dry snow), 0.75 (wet snow), 0.60 (ice); melt ponds: 0.27 (ponded ice albedo) – 0.5 (bare puddled ice); fractional surface type |
| HCLIM | 2.5 km | SICE: simple thermodynamic, 4-layer ice, 12-layer snow (ISBA-ES); ERA5: SIC, SST, sea ice initialised at 1.5m | – | snow: RACMO-based snow optical gain sizes, snow density-dependent, CROCUS-like spectral albedo; cold/melting bare ice albedo 0.71/0.61 |
| MAR | 25 km | ERA5: SIC, SST, ice initialised at 1m; thermodynamic; CROCUS snow model: 30 snow/ice levels | – | dendricity of the grains, snow thickness, the amount of meltwater (max. albedo 0.35), the presence of bare ice (max. albedo 0.5) and firn (max. albedo 0.65), cloudiness, solar zenith angle |
| RACMO | 11 km | ERA5: SIC, SST; 4-layer sea ice, fixed 1.5 m thickness, no snow | – | based on observed values (averaged over several years) of Ebert and Curry (1993), valid on 15th day of each month, interpolated otherwise |
| ERA5 | 1/4°, the northern border at the equator (∼28/11 km lat/lon direction at ISPOL) | 4-layer sea ice, fixed 1.5 m thickness, no snow | – | based on observed values (averaged over several years) of Ebert and Curry (1993), valid on 15th day of each month, interpolated othewise |

**Table 1.** Summary of relevant model characteristics for the models used in this study. Regional oceanic models: MetROMS-UHel and NEMO, regional climate models: HCLIM, MAR, and RACMO; reanalysis product: ERA5.





dynamics (Hunke and Dukowicz, 2003; Bouillon et al., 2013) with an incremental remapping scheme Lipscomb and Hunke (2004) for advection. For thermodynamics, a mushy layer formulation is used (Turner et al., 2013), with a level-pond parametrisation (Hunke et al., 2013) with refreezing based on Stefan's Law. The shortwave radiation in the atmosphere and the coupled
ice/snow layer is handled by a Delta-Eddington multiple scattering radiative transfer model (Briegleb et al., 2007).

In the model, apparent optical properties (AOPs, e.g. albedo and transmittance) of ice and snow are calculated by the Delta-Eddington model using ice and snow microscopic inherent optical properties (IOPs, i.e extinction coefficient, asymmetry parameter, single scattering albedo) that depend on snow depth, ice thickness, snow optical grain size, and ice absorption/scattering coefficients (Holland et al., 2012). It calculates the AOPs for 3 different surface types; snow-covered ice, bare ice, and
ponded ice for all thickness categories. The horizontal fraction of surface type is used to get the weighted average albedo of the grid cell. The IOPs for bare and ponded ice are taken from semi-empirical modelling based on SHEBA observations over the Arctic (Light et al., 2008), and the IOPs for snow use the equivalent ice sphere approximation for specific snow grain size from Grenfell and Warren (1999).

## 3.2   NEMO

The ocean model NEMO (Nucleus for European Modelling of the Ocean, the NEMO System Team, 2023) is coupled with the dynamic-thermodynamic sea-ice model SI3 (Sea Ice Modelling Integrated Initiative, NEMO Sea Ice Working Group, 2019). The model runs at 0.25° resolution, with a northern border at 55°S, resulting in 10 km Cartesian resolution at the ISPOL location. Oceanic forcings at the border are from the reference ORCA025 simulations (Barnier et al., 2006), while the atmospheric forcings are from the JRA55 reanalysis (coherent with the forcings of ORCA025). The grid has 121 unevenly
spaced vertical layers with a resolution of 1 m at the surface and up to 200 m at 5000 m depth.

NEMO's SI3 has 5 categories of varying thickness to represent the ice pack, with upper bounds for the categories at 0.5m, 1.1m, 2.1m and 3.7m. We note that the sea ice categories differ from those of MetROMS-UHel, as model setups were done separately, and respective sea ice model default categories were used. The last category has no upper bound and can represent arbitrary thick ice. The sea ice of any thickness category is described by two ice layers and one snow layer. The maximum sea
ice concentration per pixel is 95%, to account for unresolved polynyas and leads. The albedo of the snow/sea ice is calculated through empirical methods, taking into account factors such as ice thickness, snow depth, and surface temperature. The albedo parameterisations are based on Shine and Henderson-Sellers (1985) but revised following Brandt et al. (2005b); Grenfell and Perovich (2004). The impact of melt ponds on the albedo has also been included (Lecomte et al., 2015). Melt pond properties as given by the physical level-ice scheme characterised by the volume and level area of meltwater per unit grid cell area. They
can refreeze, in which case a lid can appear masking the effect of ponds on surface albedo.

SI3 assumes that snow, bare ice and ponded ice can coexist. However, if snow is present over sea ice, the total albedo of the grid cell is equivalent to only the snow albedo, with a deep snow asymptotic value of 0.85 for dry snow and 0.75 for melting snow (Grenfell and Perovich, 2004). If no snow is present on the ice, the total albedo is computed as a linear combination, weighted by the surface type fraction, of the bare ice albedo (which depends on ice thickness, with a maximum asymptotic



value of 0.6) and the pond albedo (ranging from 0.5 to 0.27 for deep ponds), following Lecomte et al. (2011). NEMO model
runs are available only for the years 1980 to 2018, hence for the ISPOL case and not for the Marsden field campaign.

### 3.3 HCLIM-AROME

The regional atmospheric model HARMONIE Climate (HCLIM, Belušić et al. (2020)) cycle 43 using the non-hydrostatic
HARMONIE-AROME physics package (Seity et al., 2011; Bengtsson et al., 2017) is set up for the two Antarctic domains with
2.5 km grid spacing. ERA5 reanalysis is used for atmospheric forcing. The HCLIM simulations use the Simple Sea-ice Scheme
(SICE, Batrak, 2021). SICE is a one-dimensional thermodynamic sea ice parameterisation scheme. External ice concentration
field, in this case from ERA5, is required to define ice-covered grid cells.

SICE allows for simple surface air temperature-dependent bare sea ice albedo parameterisations. In this case, cold bare ice
has an albedo of 0.71 and melting bare ice has an albedo of 0.61. When ice is covered by snow a different albedo scheme is
used. Because snow upon the ice is an insulating layer with higher albedo and lower thermal conductivity than the underlying
ice, HCLIM couples SICE, the surface modelling platform (SURFEX Masson et al., 2013), and Interactions Surface Biosphere
Atmosphere Explicit Snow 12-layer model (ISBA-ES, Boone et al., 2004; Decharme et al., 2016) to represent snow on the ice.
This is the same snow scheme used by HCLIM on land. The topmost snow layer is always less than or equal to 0.05 m thick.
The scheme describes snow thermal conductivity, radiation flux, snow accumulation, melt, and snowpack compaction in the 12
snow layers. However, there are no parameterisations of specific snow-over-ice processes, such as snow–ice formation or melt
ponds. Snow albedo is calculated using the method of CROCUS (Brun et al., 1992; Vionnet et al., 2012), in three ([0.3–0.8],
[0.8–1.5] and [1.5–2.8] $\mu$m) spectral bands, which then are combined to the total broadband albedo. The default snow albedo
depends on snow density, snow optical grain size, and ageing.

The snow grain size used in the snow albedo calculation, which is only a function of snow density, determines the reflectivity
of snow to shortwave radiation. In the default HCLIM setting, it is an empirical relation from Anderson (2008) measured in
NOAA-ARS snow research station in Danville, Vermont, USA. It is modified further by an extra ageing term for the visual
spectral band, which represents the darkening effects of snow grain metamorphism and impurities accumulation in the snow
(Vionnet et al., 2012). This term was designed for the Alps (Col de Porte) and is not representative of the Polar regions.
Therefore we updated the snow on sea ice and snow on land terms by setting the age-dependent term to zero.

The snow optical grain size and density relation of Anderson (2008) is not appropriate for the Polar regions, where snow
grain sizes are known to be smaller (Libois et al., 2015). Too large snow grain sizes lead to too low snow albedo values. This
affects surface meteorology, surface energy and mass balance. For example, the first test runs conducted for this study showed
a surface temperature warm bias as large as 5°C over sea ice compared to the ISPOL campaign measurements.

For this study, the snow grain size and density relation is updated using the output from the regional climate model RACMO
version 2.3p3 (van Dalum et al., 2019, 2020, 2022). RACMO2.3p3 has updated its spectral snow albedo and radiative transfer
scheme and the fresh snow grain sizes, which better correspond to Antarctic observations (Libois et al., 2015). The left panel
in Fig. 2 shows the updated snow grain size to snow density relation together with the HCLIM default relation. In the default
HCLIM setting, only large grain sizes are allowed, which leads to maximum snow albedo values of 0.75. With the improved




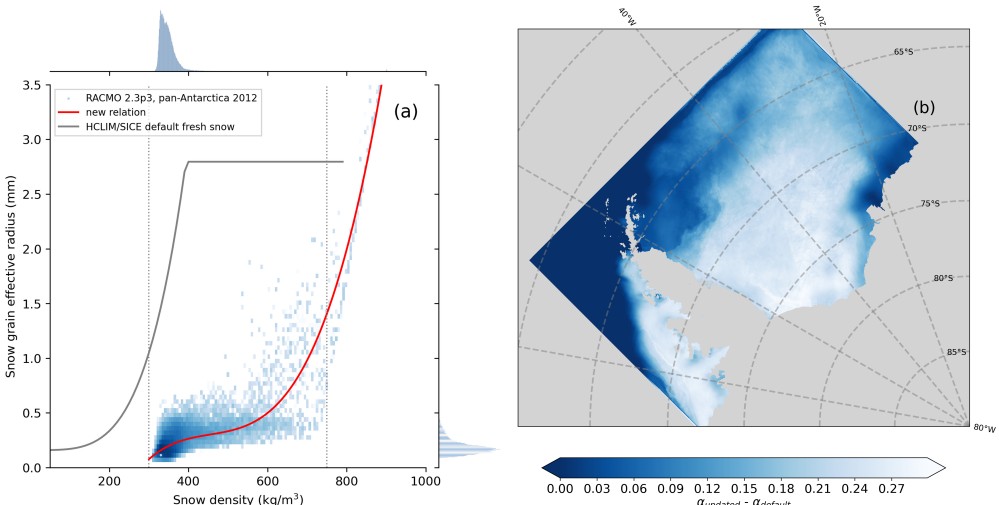

**Figure 2.** left: RACMO2.3p3 2012 yearly snow grain effective radius as a function of snow density are shown as the density plot (blue shades), fitted third order polynomial results in the updated HCLIM snow grain size to snow density relation (red solid line). The histograms show the RACMO data density. HCLIM default relation is shown with a grey solid line. The dotted lines represent the snow density limits for sea ice in SICE and on land. right: differences between HCLIM updated and the default setting mean broadband albedo for December 2004 .

grain sizes, the fresh snow albedo can reach a more realistic 0.85. The resulting differences in the monthly-averaged broadband
albedo are shown on the right panel in Fig. 2 and are as large as 0.3.

## 3.4   MAR

The Modèle Atmosphérique Régional (MAR) is a hydrostatic regional atmospheric model specifically developed for the polar areas (Gallée and Schayes, 1994). For this study, a pan-Antarctic domain with a 25 km resolution ($305 \times 272$ grid points) and 24 vertical levels, ranging from 2 meters to 17 kilometres above the surface, is used. ERA5 reanalysis is used as a lateral atmo-
spheric forcing and sea surface temperature/sea ice cover. The sea ice thickness is initialised from 1 m. Snow can accumulate at the surface of the sea ice afterwards. Sea ice can also melt to a 0.1 m minimal value.

The snow–ice module includes submodules for surface albedo, meltwater refreezing, and snow metamorphism, based on the snow model CROCUS (Brun et al., 1992; Gallee et al., 2001). Thirty levels were used to represent the snowpack over Antarctica and the surrounding sea ice. Snow albedo in the CROCUS model is calculated in three separate spectral bands as
a function of the snow properties in the top 3 cm of the snowpack. The snow albedo depends on the impurities in the snow (ageing term) and the optical diameter of snow grains. Unlike in HCLIM, the snow optical diameter takes also into account snow metamorphism, based on empirical laws describing the evolution rate of the type (dendric/non-dendric) and size of the snow grains (Brun et al., 1992). As described above for the HCLIM model, the CROCUS model was not initially made for polar regions, so the same correction was applied by setting the snow ageing term to 0. Hence, the snow albedo of MAR has



the same 3-band scheme as HCLIM, but MAR prognostically simulates the snow optical grain size, while HCLIM estimates this grain size from the snow density. If no snow is present over the sea ice, then the albedo of the bare ice varies exponentially between 0.55 and 0.5 as a function of accumulated meltwater (Lefebre et al., 2003; Tedesco et al., 2016). Finally, if the snow depth is thinner than 0.1 m, the total albedo transitions linearly between snow albedo to bare ice albedo.

When evaluating MAR albedo during the Marsden campaign, the values from the closest sea-ice gridpoint, centred at (77.79° S, 166.71° W), are used.

### 3.5 RACMOp2.4

The polar version of the Regional Atmospheric Climate Model (RACMOp2.4 van Dalum et al., 2024) consists of two major parts; the dynamical core of HIRLAM (Undén et al., 2002) that models large-scale motions in the atmosphere, and the physical parameterisations of the IFS (ECMWF, 2020), which represents physical processes on the sub-grid scale. This includes, for example, a cloud, turbulence and precipitation scheme. Additional parameterisations have been added in RACMO for a dedicated glaciated land surface tile, such as a multi-layer snow scheme, firn-densification modules, grain size evolution and melt and refreezing parameterisations.

The RACMO version used in this study, version 2.4p1, has been developed recently and includes changes in several key aspects with respect to previous model versions (van Dalum et al., 2024). The IFS physics parameterisations have been updated from cycle 33r1 (ECMWF, 2009) to cycle 47r1 (ECMWF, 2020). This includes changes in all major schemes in particular in the new cloud scheme and a new lake model is introduced. Other changes include a new multi-layer snow scheme for seasonal snow on non-glaciated surfaces and a new land-ice mask with fractional ice cover, resulting in a better representation of partly-glaciated regions. A detailed description can be found in van Dalum et al. (2024). For experiments used in this study, RACMO is run on an 11 km horizontal resolution grid and is forced by ERA5 at the lateral boundaries (van Dalum et al. submitted to the Cryosphere).

As RACMO is not coupled to an ocean model, the sea ice extent is prescribed by ERA5. RACMO uses the ECMWF IFS model for sea ice, which employs a four-layer sea ice slab with a fixed maximum thickness of 1.5 meters. There is no snow modelled on top of sea ice. The albedo is based on Ebert and Curry (1993) and thus does not depend on snow conditions. The albedo is derived by using separate values for near-infrared and visible light. The monthly albedo values from Ebert and Curry (1993) are the climatological mean values valid on the 15th of each month and linearly interpolated for other days. The effect of fresh snowfall on the albedo on sea ice during summer is therefore often not captured properly, as the albedo model does not take the actual conditions into account.

### 3.6 ERA5

Given the findings of Pohl et al. (2020); Müller et al. (2024); Batrak et al. (2024) that highlight the limitations of ERA5 sea ice albedo over the Arctic, ERA5 is included in this analysis over Antarctica. In ERA5, the sea-ice concentration is given by the Ocean and Sea Ice Satellite Application Facility (OSI SAF) (409a) dataset (from 1979 to August 2007), and the OSI SAF open dataset (September 2007 onward). Like RACMO, ERA5 also uses the ECMWF IFS model for sea ice. ERA5 considers



the Marsden CS to be on land, not on sea ice. When evaluating ERA5 albedo during the Marsden campaign, the values from the closest sea-ice gridpoint, centred at (77.85° S, 166.54° W), are used. Since HCLIM, MAR, and RACMO use sea ice

concentrations from ERA5, it can be assumed that these models extrapolate the sea ice concentrations from adjacent ERA5 ocean grid points to areas classified as land in ERA5.

## 4 Results

### 4.1 Atmospheric conditions

The model's ability to accurately reproduce top-layer snow and ice conditions is influenced by atmospheric modelling (as

in HCLIM, RACMO and MAR) or by the atmospheric forcing (as in MetROMS-UHel and NEMO with ERA5), particularly during the melt season. The comparison between modelled and observed 2 m temperature is presented in Fig. 3, while additional atmospheric variables such as air pressure, relative humidity, wind direction, and wind velocity are provided in Figs. A1 and A2.

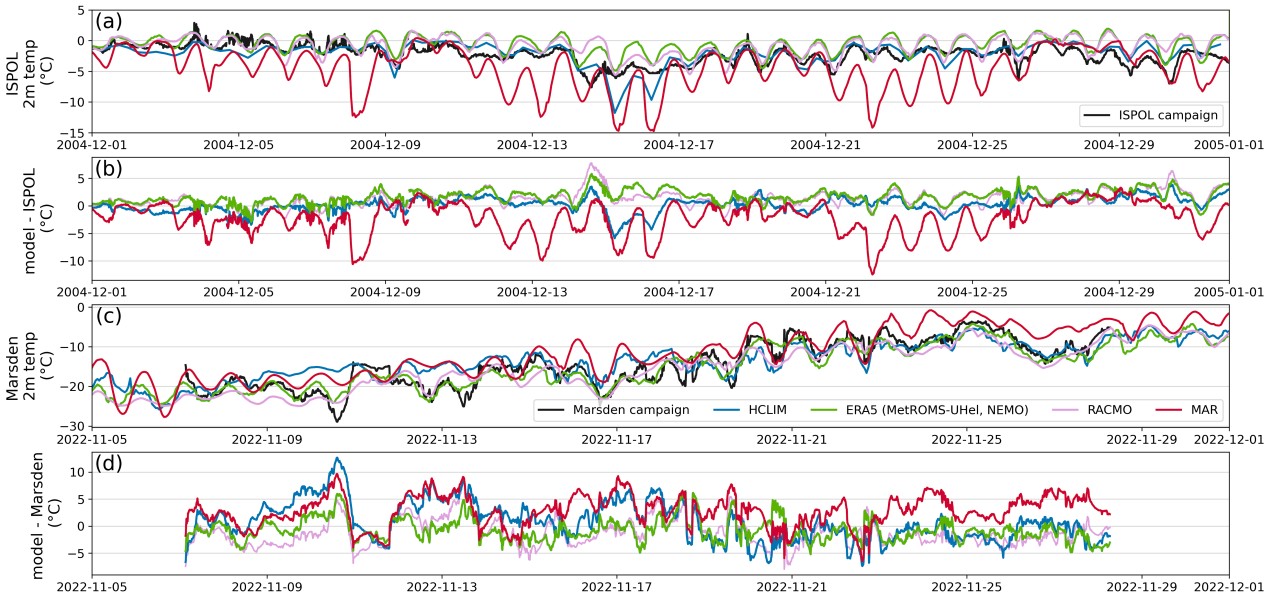

**Figure 3.** Modelled and observed 2m temperature, and the temperature differences during the ISPOL campaign (panels a-b) and Marsden campaign (c-d).

During the ISPOL campaign, the weather was warm for this location, with the air temperature mostly above -5°C and

295 even around zero degrees during the first week of December. HCLIM reproduces the surface temperature well (with a mean difference of 0.2°C), as well as the surface pressure, wind speed and direction. RACMO and ERA5 have, on average, a 1.4°C and 1.6°C warm bias respectively, indicative of the low sea ice albedo, as discussed later. ERA5, with a resolution of 0.25°





(approximately 28 km), exhibits a temperature pattern comparable to the 11-km resolution RACMO. In contrast, MAR, which has the coarsest spatial resolution of the RCMs in the set at 25 km, displays significant diurnal variability and is, on average, 2.3°C colder than observations, with deviations reaching up to 13°C.

The conditions during the Marsden campaign were notably different – the air temperatures were well below freezing (mostly below -10°C). The Marsden field campaign took place in a challenging region for models, located a few kilometres north of Ross Island (see Fig. 1). Notably, in both ERA5 and MAR, the Marsden field campaign site is inaccurately represented as land rather than sea ice. Even so, HCLIM, RACMO, and ERA5 differ by only 0.8, 1.2, 0.8°C from the observations. Figure 3 shows that on average, MAR exhibits a 2.8°C warm bias and pronounced diurnal temperature fluctuations. Figure A2 reveals the problems that atmospheric models have in simulating the correct wind direction over the Marsden campaign site: instead of the observed predominantly southerly winds, the models predicted more easterly winds. The reason for this discrepancy is the resolution differences and the proximity of Ross Island and the Ross Ice Shelf.

## 4.2 Comparison between modelled and measured sea ice albedo

In-situ measurements provide a detailed understanding of the temporal variability of sea ice albedo, along with general snow and ice characteristics. However, these measurements fall within the sub-grid scale of the models used in this study, with HCLIM offering the highest resolution at 2.5 km. Additionally, both campaign locations present challenges for model reproduction – McMurdo Sound is characterised by complex topography, and the ISPOL campaign took place close to the edge of the modelled December sea ice extent. Figure 4 shows the month-long ice and/or snow albedo time series. Table 2 presents the mean albedo values with standard deviation. Since both campaigns measured either on an ice floe or land-fast ice, we compare these measurements to the modelled sea ice albedo rather than the modelled total albedo, which would include open water albedo if present in a model grid cell.

| Campaign | Observed (snow/ice) albedo | MetROMS-UHel | NEMO | HCLIM | MAR | RACMO2.4 | ERA5 |
|---|---|---|---|---|---|---|---|
| ISPOL | 0.78 (0.06) (FYI) 0.82 (0.05) (SYI) | 0.78 (0.08) | 0.81 (0.02) | 0.78 (0.03) snow 0.64 (0.05) ice | 0.83 (0.01) | 0.69 (0.06) | 0.63 (0.04) |
| Marsden | 0.79 (0.03) (CS) | 0.85 (0.00) | - | 0.84 (0.00) snow 0.71 (0.00) ice | 0.74 (0.13) | 0.78 (0.03) | 0.82 (0.03) |

**Table 2.** Average observed and modelled snow/ice albedo values and corresponding temporal standard deviations due to variability (in brackets) over the campaign period. ISPOL campaign site: measurements done at the FYI and SYI radiation sites. Marsden campaign: measurements conducted at the fixed radiation station at the camp site (CS). NEMO model runs are available only for the ISPOL case and not for the Marsden field campaign.

On the ice floe, to which RV Polarstern was moored during the ISPOL campaign, snow depth decreased on average from 32 cm to 14 cm on FYI, and from 95 to 83 on SYI during December 2004 (Nicolaus et al., 2009). On the FYI, the thinner snowpack had a mean albedo of 0.78 (std = 0.06), while on SYI the thicker snow cover had a mean albedo of 0.82 (std = 0.05). The standard deviation of albedo primarily reflects the diurnal variability observed during the ISPOL campaign, driven





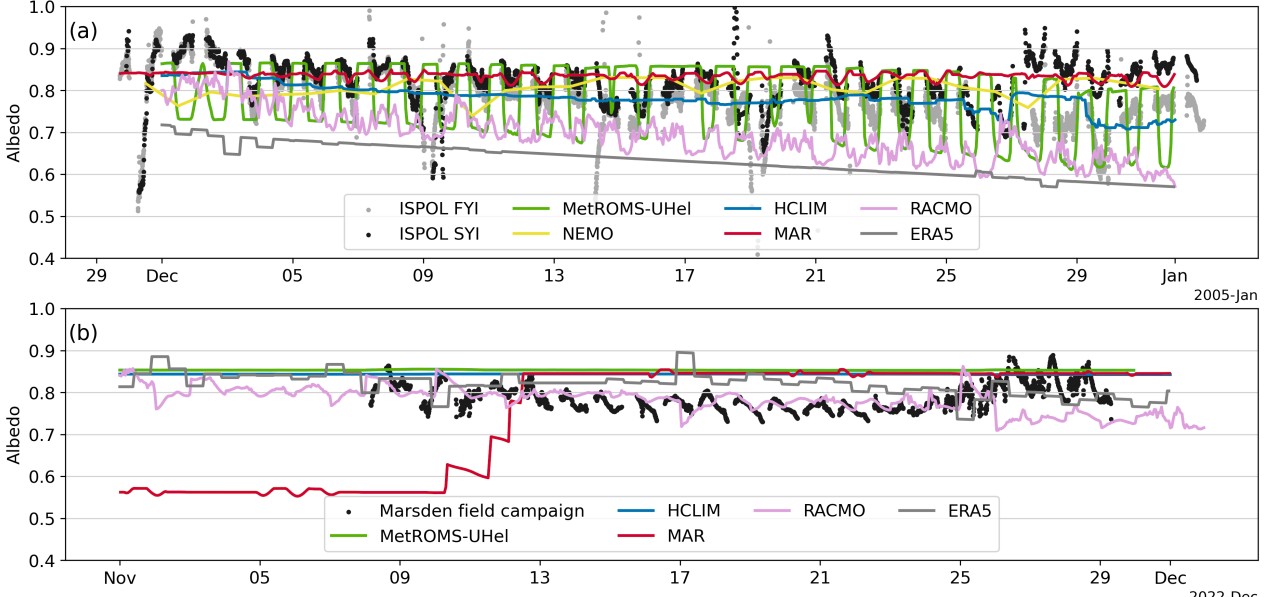

**Figure 4.** Observed and modelled sea ice albedo time series for the ISPOL (panel a) and the Marsden (b) campaigns. Some models, such as MetROMS-UHel and HCLIM, output separate albedo values for snow and ice. Others, like MAR, NEMO, and RACMO2.4, provide a grid-box-averaged albedo that combines contributions from ocean, ice, and snow. For these models, we derived the snow and ice albedo by isolating the ocean contribution using sea ice concentration data.

by near-melt snow conditions during the day, refreezing at night, and changes in the solar zenith angle. Additional changes in observed albedo were influenced by snowfall, drifting snow, and the relocation of the equipment (Vihma et al., 2009).

In the McMurdo Sound, the 1 to 4 m thick land-fast ice had reportedly thin, or patches of thin (∼0.2 m) snow on top.
Overall, a mean surface albedo of 0.79 (std = 0.03) was observed. A snowstorm during the last week of the campaign led to an increase in albedo which reached up to 0.9. The observed albedo was primarily influenced by variations in drifting snow accumulation patterns within the field of view of the downward-looking pyranometer. The snow was often reported as optically translucent, allowing the bare ice beneath to be visible. In nature, bare ice albedo varies between 0.1 and 0.7 depending on the ice thickness, and the presence of cracks, melt, brine pockets and air bubbles (Allison et al., 1993; Brandt et al., 2005a).
Bare ice albedo values above 0.56, however, have been observed only during melting, when the surface of the ice decomposes into white ice or "surface scattering layer". White ice is typical of summer Arctic sea ice but has never been observed over Antarctic sea ice (Grenfell and Maykut, 1977). However, Traversa and Di Mauro (2024) recently reported the presence of this "surface scattering layer", which they referred to as the weathering crust, on over blue ice areas of ice shelves of the Northern Victoria Land, Antarctica.
Considering that the measurements are from a single point location while model output represents grid cell averages, the models reproduce the high albedo values and diurnal variability at the ISPOL location but struggle to accurately capture the





albedo variability at the Marsden site. MetROMS-UHel successfully captures both the overall level of albedo and the strong diurnal variability observed during the ISPOL case, with a mean modelled albedo of 0.78 (std = 0.08). In the Marsden case, MetROMS-UHel suggests a constant albedo of 0.85. NEMO output was available only for the ISPOL case study, in which case

it has a monthly mean albedo of the sea ice of 0.81 (std = 0.02). As only daily NEMO albedo fields were saved, any modelled diurnal variability is lost for this analysis.

HCLIM simulates a snow albedo of 0.78 (std = 0.03) for the ISPOL case, close to the observed albedo over SYI. The HCLIM snow albedo in the Marsden campaign case is 0.84 (std = 0.00). The HCLIM bare ice albedo for the Marsden case is 0.71 (std = 0.00), coming from the simplistic two-value (0.61 or 0.71) parameterisations of bare ice albedo in the model. HCLIM bare

ice albedo is too high, as bare ice albedo values above 0.56 are unlikely in this context.

MAR simulates a monthly mean albedo of 0.83 (std = 0.01) and 0.74 (std = 0.13) for the ISPOL and Marsden field campaigns respectively. A small diurnal variation is present in the model output at the ISPOL location, but not to the extent of the observations or of the MetROMS-UHel simulations. A significant increase in MAR's surface albedo, from 0.55 to 0.85, occurred between the 10th and 13th of November when a thin layer of snow fell within the previously bare ice grid cell in the

model, resulting in an immediate rise in the modelled albedo.

RACMO and ERA5 both use time-interpolated monthly albedo values based on the albedo scheme developed by Ebert and Curry (1993) for the Arctic sea ice. Differences between the two models come from the resolution: RACMO was run at 11 km resolution, but ERA5 has a 0.25° (∼28 km) resolution. At the ISPOL location, both RACMO and ERA5 had lower surface albedo values than expected from observations, with RACMO at 0.69 (std = 0.06) and ERA5 at 0.63 (std = 0.04). However,

over the Marsden campaign site, RACMO more accurately represents the observed conditions compared to all other models. In this instance, the assigned monthly average sea ice albedo values of Ebert and Curry (1993) align with the albedos observed in this case study. ERA5 differs in this case, as the values from the nearest sea-ice grid point are used because the campaign site is inaccurately represented as land rather than sea ice.

## 4.3  Comparison between modelled and measured sea ice and snow thickness

| Campaign | Observed (snow/ice) depth (m) | MetROMS-UHel | NEMO | HCLIM | MAR | RACMO2.4 | ERA5 |
|---|---|---|---|---|---|---|---|
| ISPOL snow: | 0.23 (FYI) / 0.51 (SYI) | 0.14 (0.08) | 0.71 (0.09) | 0.58 (0.13) | - | - | - |
| ice: | 0.95 (FYI)/ 0.80–1.60 (SYI) | 1.15 (0.04) | 3.53 (0.31) | 1.85 (0.06) | snow+ice: 1.96 (0.05) | 1.5 | 1.5 |
| Marsden snow, CS: | 0.21 | 0.09 (0.00) | - | 0.15 (0.00) | - | - | - |
| other sites: | 0.30 (RS), 0.02 (NIS), 0.14 (TS), 0.005 (TRS) | | | | | | |
| ice, CS: | 2.31 | 1.73 (0.03) | - | 1.97 (0.1) | snow+ice: 0.52 (0.01) | 1.5 | 1.5 |
| other sites: | 2.71 (RS) 1.18 (NIS), 2.84 (TS), 1.09 (TRS) | | | | | | |

**Table 3.** Measured in-situ and grid-box averaged modelled snow and ice thicknesses and corresponding temporal standard deviations due to variability (in brackets) over the campaign period. ISPOL campaign: measurements over FYI and SYI. Marsden campaign: measurements from camp site, but also from Ridge Site (RS), Transition Site (TS), New Ice Site (NIS), and Turtle Rock Site (TRS) shown in Fig. 1.





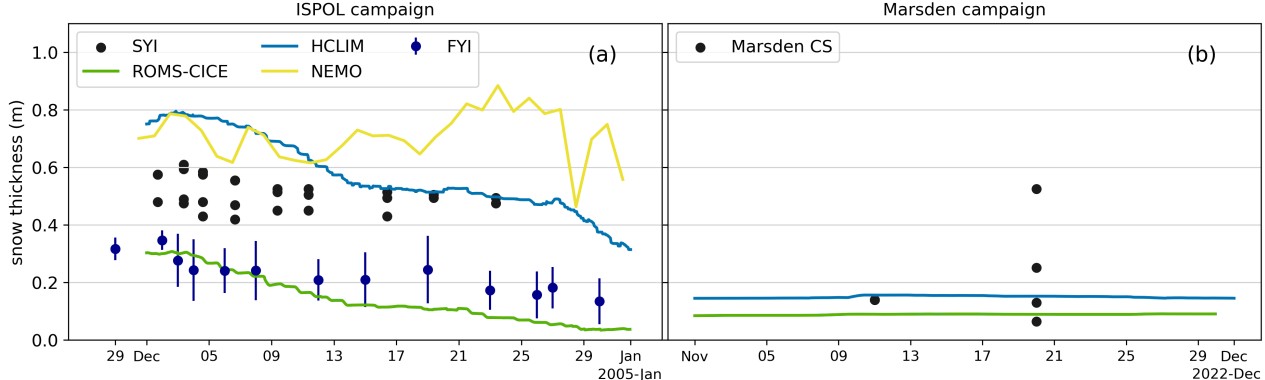

**Figure 5.** Snow thickness over sea ice time series for the ISPOL (panel a) and the Marsden campaigns (b). ISPOL campaign: Measurements which were taken closest to the FYI and SYI radiation sites. Marsden campaign: Measurements conducted near the fixed radiation station at the camp site (CS). For the models, the grid-box average snow thickness is shown if available. All the other models provide hourly output, whereas NEMO provides daily output. ERA5 and RACMO do not have snow on sea ice. MAR provides combined snow and ice thickness and is excluded here.

A certain optical thickness of snow is required to obscure the underlying ice, and similarly, bare ice must reach a specific thickness to mask the darker ocean beneath. Therefore, to interpret the comparison between modeled and observed albedo it is valuable to analyse the modelled snow and ice thickness in comparison to the observed values. During both campaigns, ice and snow thicknesses were measured at multiple locations across the campaign sites, leading to broad distributions, especially of snow thicknesses across both sea ice campaign regions. Only NEMO and MetROMS-UHel have sub-grid ice categories, which

are used to analyse such unresolved spatial variability of sea ice thickness in Sect. 4.4. Here, modelled grid-box average sea ice thickness and snow thickness are examined and compared with measurements taken closest to the location of the albedo measurements. The models are not regridded to a common low-resolution grid in order to preserve high-resolution information.

     Figure 5 and Table 3 compare the measured snow thickness in the ISPOL and Marsden campaigns with grid-averaged model outputs. Generally, the thicker the sea ice, the greater the ice freeboard and the amount of snow it can support.

The ISPOL location measurements were done over the FYI and SYI. MetROMS-UHel snow thickness follows the level and trend of the ISPOL FYI measurement site: 0.35 m thickness at the beginning of the month, and only a few cm thick at the end. The grid-box average bare sea ice thickness underneath is 1.15 m. HCLIM predicts thicker snow, 0.8 m at the beginning of the month, dropping to 0.4 m at the end of the month, with a 1.85 m sea ice underneath the snow. NEMO also predicts a thicker, 0.71 m, snow cover over a 3.5 m bare sea ice. While HCLIM shows a decrease in snow thickness over the month, NEMO

snow thickness is variable, but not decreasing. The snow cover modelled by NEMO and HCLIM is thicker than measured at the ISPOL SYI and FYI locations. However, there were sites at the ISPOL ice floe where snow thicknesses were in that range (Nicolaus et al., 2009). MAR provides only the combined sea ice and snow depth output, which amounted to 1.96 m. RACMO has a fixed sea ice thickness of 1.5 m and no explicitly modelled snow on top (Table 1).




During the Marsden campaign, snow thickness measurements were done in multiple locations (averages given in Table 3).
Reportedly, the thicker ice at CS was covered by a thin, 0.21 m, but heterogeneous snow layer with small spots (1-2 m$^2$) of bare ice. The thinner ice (at NIS and TRS) was mostly bare with small patches (1-2 m$^2$) of thin (approximately 1–2 cm) snow (Dadic 2025, in prep.). In Fig. 5, the model output is exclusively compared to the measurements at CS, closest to the radiation time-series measurements. The models predict a thin, 0.09 m (MetROMS-UHel) or 0.15 m (HCLIM), uniform snow cover. The thickness of bare sea ice is modelled to be 1.73 and 1.97 m for MetROMS-UHel and HCLIM respectively. Because MAR
considers the Marsden CS to be on land, and the closest sea-ice gridpoint is further away, MAR's snow and sea-ice thickness are not fully representative. MAR's combined sea ice and snow depth is 0.50 m at the start of the period, and increases to 0.53 m throughout 10-12th November, causing the increase of albedo seen in Figure 4 due to snowfall. ERA5 and RACMO sea ice have a constant and uniform thickness of 1.5 m, and no modelled snow.

### 4.4   Sub-grid sea ice characteristics around the Marsden campaign site

Even within a small sea ice region, multiple types of sea ice can be present. NEMO and MetROMS-UHel have sub-grid ice categories, which can be used to analyse the unresolved spatial variability of sea ice thickness within each model grid cell. However, NEMO output for the Ross Sea in November 2022 was unavailable, as the model data extends only up to 2018. Therefore, data from November 2004 for the same region is used instead. The model's intrinsic bare ice characteristics per category remain consistent year to year, while the presence of ice types, the fractional snow cover, and the cloud coverage
change.

Figure 6a-b shows the probability distributions of the albedo measured from a drone flying vertical profiles at an altitude between 30 and 50 m over thick ice (∼2.1 m, CS) and between 30 m to 70 m altitude over thin ice (∼1.2 m, NIS) with patchy snow cover. At these heights, the footprint of the drone's downward-facing pyranometer, calculated as the area contributing to 90% of the received radiation, has a radius ranging from 90 m (at 30 m height) and 210 m (at 70 m height). Hence, these
probability distributions represent the albedo averaged over footprints of ∼0.02-0.04 km$^2$ and, therefore, are better suited for validating satellite albedo products and model simulations than the fixed station measurements.

Over the thick ice, the downward-facing pyranometer of the fixed station mostly receives the radiation reflected from the surface type located just underneath it (∼a few m$^2$), having a mean albedo of 0.80. In contrast, the pyranometer on the drone captures radiation from all surface types within its footprint, weighted by their respective fractions. This resulted in a higher
mean albedo of 0.86. Since the fraction of bare ice was significantly smaller than that of snow-covered ice, the drone measured more radiation reflected from the patchy snow cover compared to the bare ice.

The spread of the drone-based albedo probability distributions, which represent the measurement uncertainty during ∼10 minute flight, is much narrower than the spread of the probability distribution of the albedo measured from the fixed station close to the surface during a one-month-long observation period (Figure 6b). The temporal variability of the fixed station
albedo is mostly due to changes in snow thickness caused by the continuous snow drift, erosion and formation of dunes, and not by snow metamorphism (because the surface temperature was well below freezing for the whole period). Hence, we can



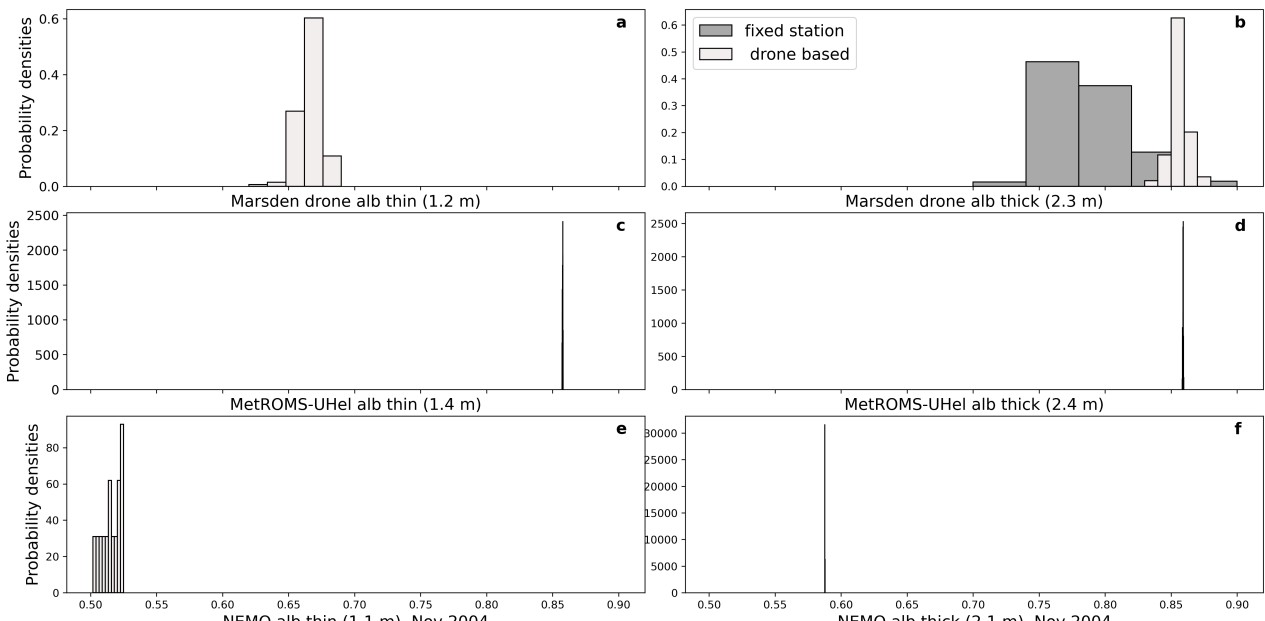

**Figure 6.** Probability distributions of drone-based or fixed station sea ice albedos at the Marsden field campaign (a,b), measured over thinner ice (1.2 m; a) and thicker ice (2.3 m; b), and the modelled MetROMS-UHel albedos (c,d) for closest thickness categories (1.4 m and 2.4 m respectively), as well as the modelled NEMO albedos (e,f) for 1.1 m, and 2.1 m ice categories.

argue that it also represents the spatial albedo variability, though biased toward the albedo of the most frequent surface type that happened to occur right below the pyranometer.

Drone-based albedo measurements over the thin ice showed a mean albedo of 0.66, as the ice surface was dominated by bare
ice with snow patches. The thinner and younger ice was formed in August 2022, which is why it had less snow than the thicker ice, which was formed in March 2022.

The modelled MetROMS-UHel albedos for the 1.4 m and 2.4 m ice thickness categories, and the modelled NEMO albedos for 1.1 m, and 2.1 m ice categories are included in Fig. 6c-d and Fig. 6e-f, respectively. The selected model thickness categories are the closest to the observed sea ice thicknesses.

The discrepancy between the observations and the models over the thinner ice is large: $\Delta_{\mathrm{mean}} = 0.2$, and 0.14 for MetROMS-UHel and NEMO respectively. In this case, the MetROMS-UHel modelled albedo is snow-dominated, as both ice categories are covered with a layer of snow. NEMO had snow-free conditions in November 2004. NEMO's bare ice albedo depends mainly on ice thickness, with a maximum albedo value of 0.5 for 1 m thick sea ice, and approaching an albedo value of 0.6 for 1.5 m and thicker ice, with additional adjustments based on cloud fraction.

The drone-based albedo measurements over the thicker ice are reproduced by MetROMS-UHel, with differences between the average values of the two distributions being negligible ($\Delta = 0.004$). MetROMS-UHel albedo has a narrower distribution




the model to bare ice albedo but is far too low compared to the observations.

These model-to-observation comparisons demonstrate that direct comparisons with surface-based point measurements are
not meaningful when there is meter-scale spatial heterogeneity in surface albedo. MetROMS-UHel and NEMO have informa-
tion on sub-grid ice categories, but the snow on the ice, if any, covers it uniformly. The comparisons also show that the correct
estimate of the snow cover depth is essential to correctly model the sea-ice albedo.

## 4.5 Spatial albedo variability over the McMurdo Sound area

Expanding the analysis to cover larger areas that align with the model resolution is a fairer evaluation of the models. This
section compares the model output to high-resolution (30 m Landsat 9 and 20 m Sentinel 2) satellite images over a limited area
($260 \times 260$ km$^2$ Landsat 9 and $110 \times 110$ km$^2$ Sentinel 2) in McMurdo Sound, which includes the Marsden campaign site. In
Section 4.6, the comparison is expanded to lower-resolution satellite observations over larger areas, covering the entire Ross
and Weddell Seas.

Landsat 9 albedo observations on the 1st of November in Fig. 7, and Sentinel 2 observations on the 14th of November, 2022
in Fig. 8, are compared to the modelled total albedo of sea ice (combination of ice, snow and ocean). The satellite images show
a variety of albedo values over the region: open ocean with an albedo of around 0.1, thicker bare land-fast ice with an albedo
of 0.7 and higher depending on the snow cover, and everything in between. The image also shows bare and snow-covered land.

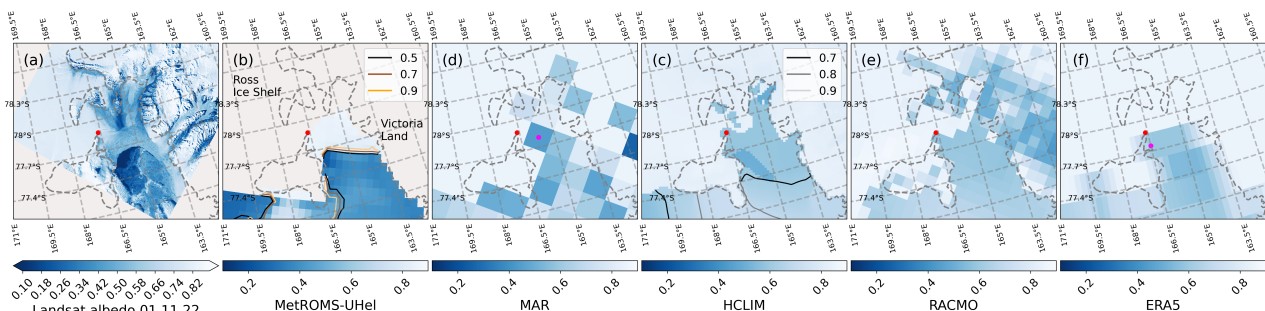

**Figure 7.** Landsat 9 albedo over the Marsden campaign site (panel a), on the 1st of November 2022 compared with the modelled albedo,
closest hourly output to the satellite flyover, of MetROMS-UHel (b), MAR (c), HCLIM (d), and RACMO (e), and ERA5 reanalysis (f). For
these maps and the ones below, where relevant, we use combined bare ice or snow and ocean albedo (with a value of 0.06) using sea ice
concentration for scaling; for MetROMS-UHel, we use the total snow and ice albedo and combine with ocean albedo similarly; for MAR,
we used albedo averaged across all surface types. Snow fraction contours (0.5,0.7,0.9 with black, brown and yellow lines) are shown atop
MetROMS-UHel. Sea ice concentration contours (0.7,0.8,0.9 with black, grey and light grey lines) are shown atop HCLIM, which seem to
describe the spatial variability of the albedo better. Red points indicate the location of the Marsden field campaign site, and the magenta
points are the locations where the model values were taken.





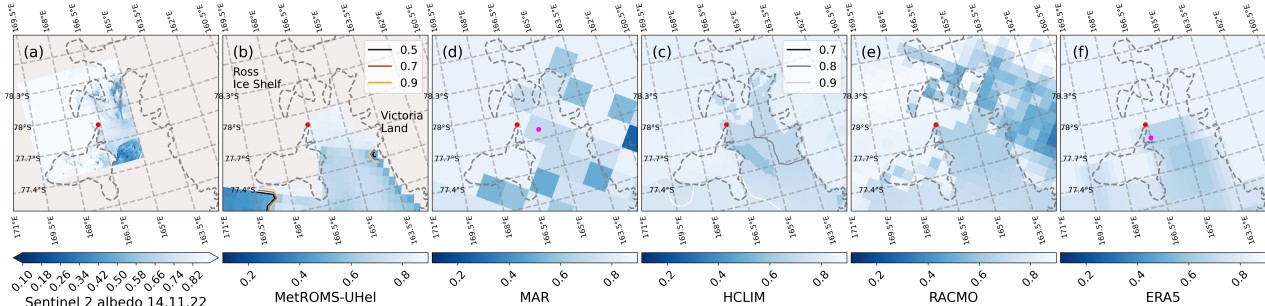

**Figure 8.** Sentinel 2 albedo over the Marsden campaign site, on the 14th of November 2022 and the modelled total albedos, closest hourly output to the satellite flyover, of MetROMS-UHel (b), MAR (c), HCLIM (d), and RACMO (e), and ERA5 reanalysis (f). Snow fraction contours (0.5,0.7,0.9 with black, brown and yellow lines) are shown atop MetROMS-UHel, but sea ice concentration contours (0.7,0.8,0.9 with white, grey and black lines) are shown atop HCLIM, which seem to describe the spatial variability of the albedo better.

Overall, the albedo in the region increased between the two dates. The sea ice concentrations in the McMurdo Sound locally increased, which explains the albedo increase over the sea ice. However, the albedo over land is about 0.06 higher
in the Sentinel 2 image compared to the Landsat 9 image. The higher albedo over land on the latter date could have various causes, such as solar zenith angle differences due to satellite fly-by times, using different satellite sensors and data processing variations. The uncertainty of the Landsat 9 albedo product, based on the applied methodology described in Sect. 2.3, is ±0.02 in polar regions (Traversa et al., 2021). For Sentinel 2 albedo imagery, an uncertainty of ± 0.05 was estimated by Naegeli et al. (2017); Di Mauro et al. (2024).

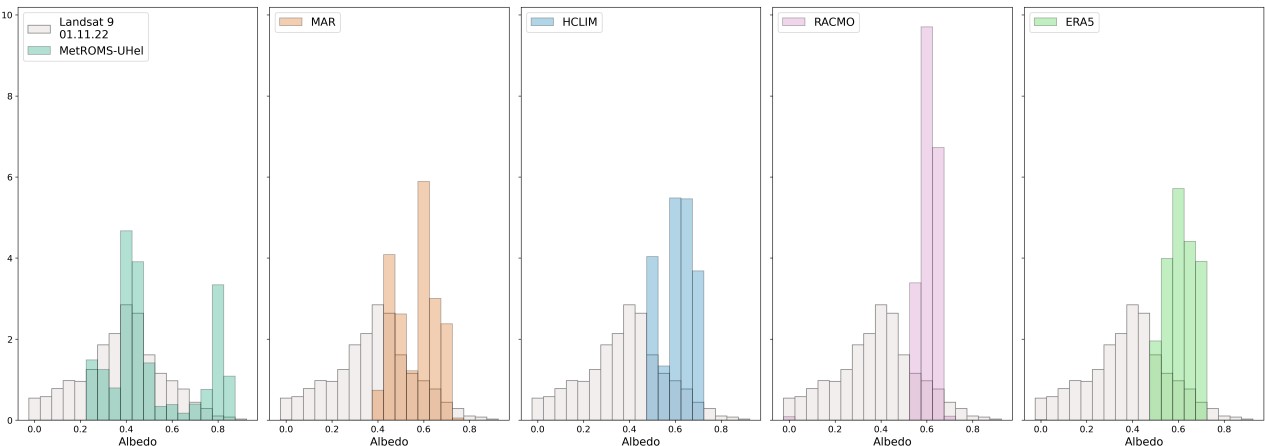

**Figure 9.** Sea ice albedo distribution (land removed) over the Landsat 9 image domain.

Because of the small area of the sea ice in the satellite images, more quantitative comparisons are difficult. Figure 9 shows the probability distributions for the sea ice albedo on the Landsat 9 image (Fig. 7), highlighting model deficiencies over the limited





sample area. The surface albedo over the ocean/sea ice part of the Landsat 9 image has a wide distribution between 0 (open ocean) and 0.9 (snow-covered sea ice) and a clear maximum at 0.4. The regional climate models, particularly MAR with its low spatial resolution, lack detailed representation of the observed surface albedo. RACMO, ERA5 and HCLIM have a narrow

distribution of sea ice albedo values, between the 0.4-0.75 range. Although HCLIM operates at a high, 2.5 km resolution, its sea surface temperature and sea ice concentration fields are derived from the 0.25° (~28 km) resolution ERA5 reanalysis. As a result, any high-resolution sea ice variables output by HCLIM are influenced by the lower-resolution ERA5 forcing. The same is true for RACMO and MAR.

Among the tested models, MetROMS-UHel displays the largest variation in sea ice albedo over the same area covered by

the satellite albedo products. The distribution of the modelled sea ice albedo is bimodal. Over this small sea ice area, the sea ice concentration in the model remains generally high (above 90%). The fractional snow cover on top of the sea ice is also a key factor in determining sea ice albedo. The combination of high sea ice concentration (>95%) and snow fraction (>90%) in McMurdo Sound resemble the features of land-fast sea ice observed in the satellite images. This results in the high albedo peak centred at 0.8 in the MetROMS-UHel albedo distribution. However, unlike the Landsat 9 image, the MetROMS-

UHel output fails to capture the distinct characteristics of free-floating sea ice both spatially and in its probability distribution. The MetROMS-UHel sea ice albedo distribution exhibits a lower albedo peak centred at 0.4, but albedos below 0.3 are not represented.

Model comparisons with such high-resolution, albeit limited-area, satellite products are valuable and should be further explored, ideally using larger satellite image datasets. These comparisons reveal the complexities of observed sea ice, highlighting

model deficiencies while guiding advancements in model development.

## 4.6   Spatial albedo variability over the Weddell and Ross Seas

The largest area of comparison is the Weddell (2 799 169 km$^2$) and Ross Seas (3 200 661 km$^2$), defined by the size of the HCLIM respective domains. The maps for the satellite images, the model and ERA5 over the Weddell Sea and Ross Sea are shown in Figs. 10 and 11. The distributions of the surface albedo over these areas are shown in Figs. 12 and 13. Furthermore,

similar distributions, but for sea ice and snow only (with ocean albedo removed) are shown in Figs. B1 and B2 in Appendix B, illustrating the snow and ice albedo parameterisation distribution, without sea ice concentration effects.

The surface albedo in the sea ice zone is always strongly influenced by sea ice concentration. HCLIM, MAR and RACMO use the sea ice concentration fields from ERA5, which itself is satellite-derived. The spatial surface albedo patterns in Fig. 10 are therefore the same in these models. However, RACMO and ERA5 have lower sea ice albedo compared to CLARA-A3

and other models. This is better revealed in Fig. 12. The sea ice-zone albedo in ERA5 has the first mode at 0.60, although the maximum observed sea ice-zone albedo is as high as 0.85. The secondary mode of 0.05 is from points with no sea ice cover. RACMO has the first mode at 0.65. This peak does not come from sea ice concentrations in the area but from the sea ice albedo parameterisation. Figure B1 shows that pure snow and ice parameterisation over the whole region is about 0.65 for both ERA5 and RACMO.





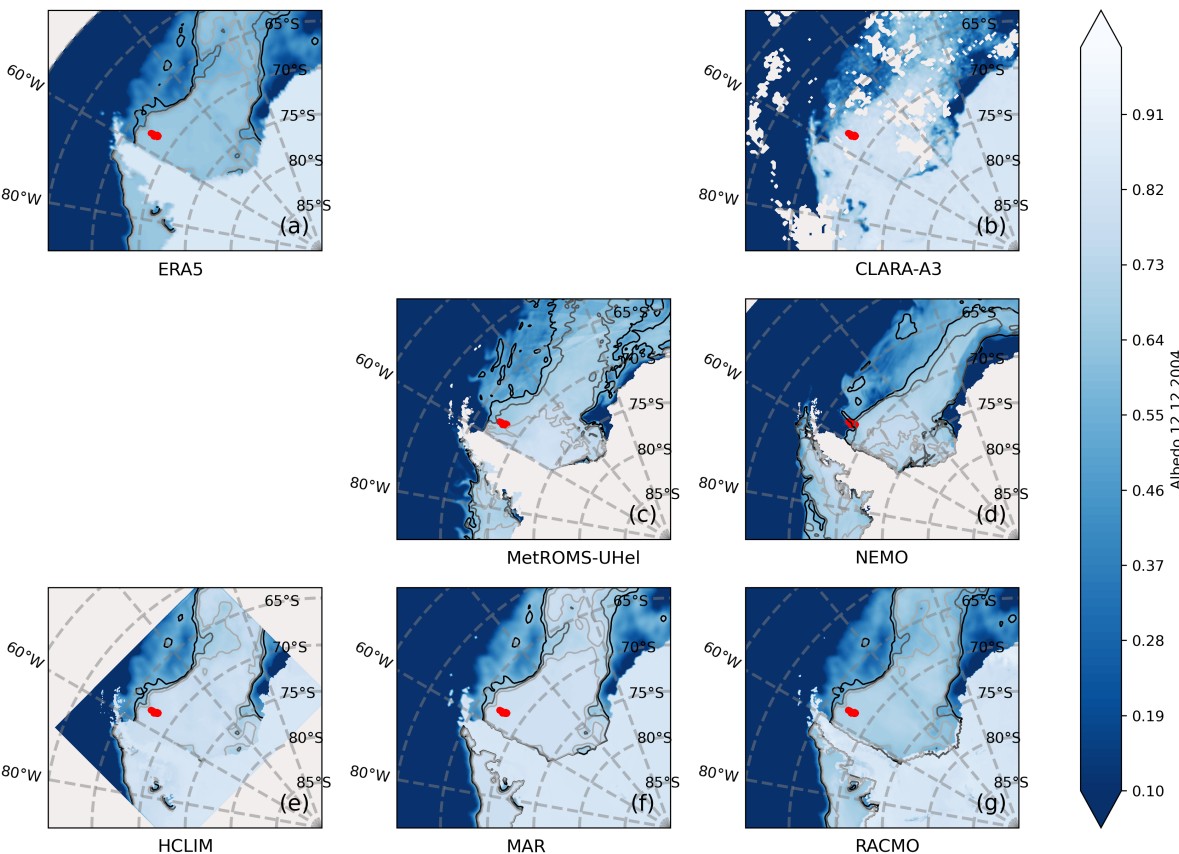

**Figure 10.** ERA5 (a) reanalysis and CLARA-A3 (b) satellite albedo products from 12.12.2004 over the Weddell Sea domain as reference for model validation, and corresponding daily mean albedo maps from MetROMS-UHel (c), NEMO (d), HCLIM (e), MAR (f) and RACMO (g) models. Sea ice concentration contours are shown on top with black, grey and light-grey contour lines for the values of 0.7, 0.8, and 0.9 respectively. The location of the ISPOL field campaign is marked with red.

The regional oceanic models differ in the sea ice concentration patterns. We note that NEMO predicts the sea ice margin to be close to the ISPOL field campaign site. NEMO also has the first mode lower than expected from the observations, at 0.70, and does not describe 0.80 and higher albedo values due to overall lower sea ice concentrations. Figure B1 shows that the first mode for snow and ice albedo is at 0.80. Therefore, the main difference in the surface albedo comes from sea ice concentrations in NEMO and MetROMS-UHel, not from their albedo parameterisations. The spatial distribution and density distribution of 490  the observed albedo are best reproduced by the MetROMS-UHel model.

    Over the Ross Sea domain, shown in Fig 11, the albedo spatial pattern is a combination of sea ice concentration and snow fraction. The sea ice concentrations in the defined domain are mostly high and we see open water only at the Ross Sea polynya. CLARA-3A image shows more small-scale variability, but the distribution shows a clear mode at 0.75-0.80. The models predict it correctly but do not describe the variety of lower, 0.2 - 0.7 albedo values. Exceptions are the ERA5, with a





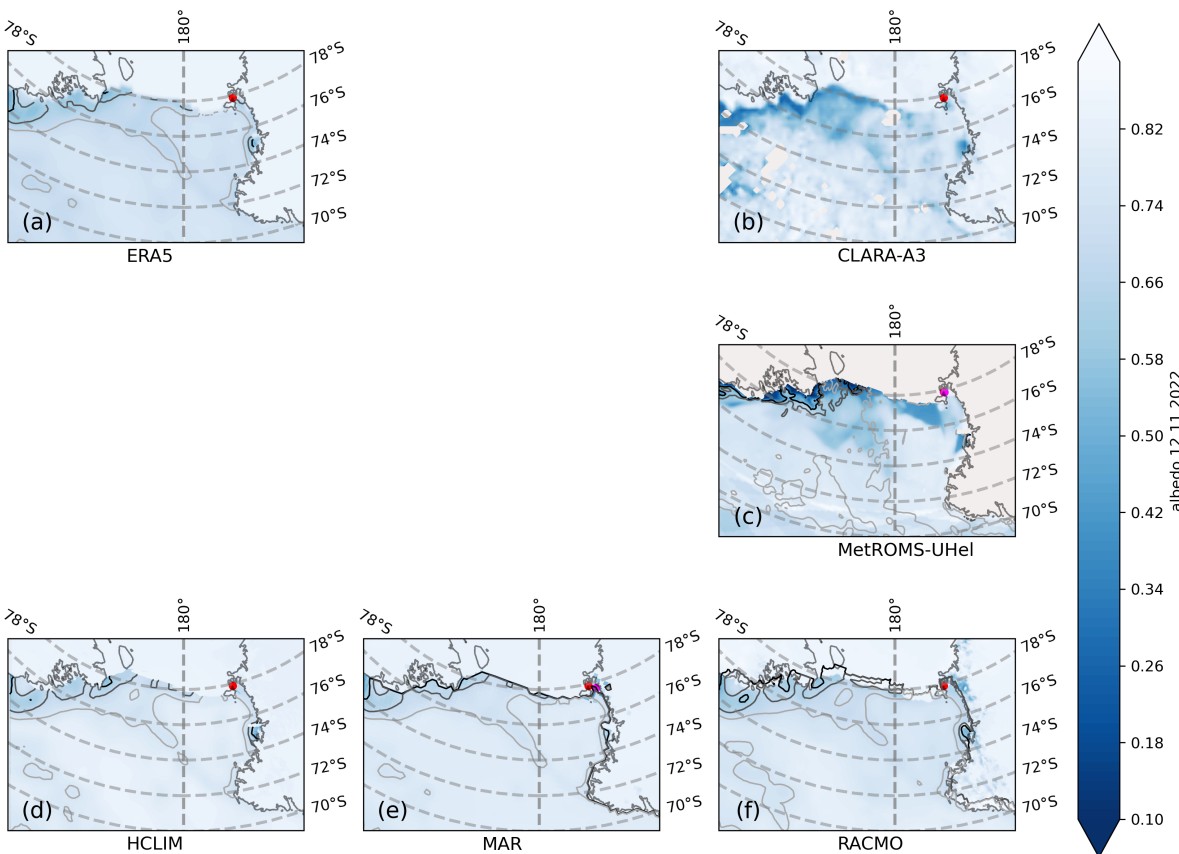

**Figure 11.** ERA5 reanalysis (panel a) and CLARA-A3 satellite albedo (b) products from 12.12.2004 over the Ross Sea domain as reference for model validation, and corresponding daily mean albedo maps from MetROMS-UHel (c), HCLIM (d), MAR (e) and RACMO (f) models. Sea ice concentration contours are shown on top with black, grey and light-grey contour lines for the values of 0.7, 0.8, and 0.9 respectively. The location of the ISPOL field campaign is marked with red. NEMO model runs are available only for the ISPOL case and not for the Marsden field campaign.

lower-than-observed strong mode at 0.7, and MetROMS-UHel, which matches the distribution of observed albedo values well. Notably, MetROMS-UHel reproduces the coastal polynya in front of the Ross Ice Shelf that can be seen from CLARA-A3 in Fig. 11.

RACMO does better over the Ross Sea in November 2022 than over the Weddell Sea in December 2004. When interpolating between monthly climatological averages in the sea ice albedo parameterisation of Ebert and Curry (1993), the predictions are occasionally accurate. The regional climate models MAR, HCLIM and RACMO have similar behaviour over the Ross Sea, although RACMO has slightly lower first-mode snow albedo. The small differences come from parameterisations only, as the sea ice concentration fields are all the same ERA5 input.





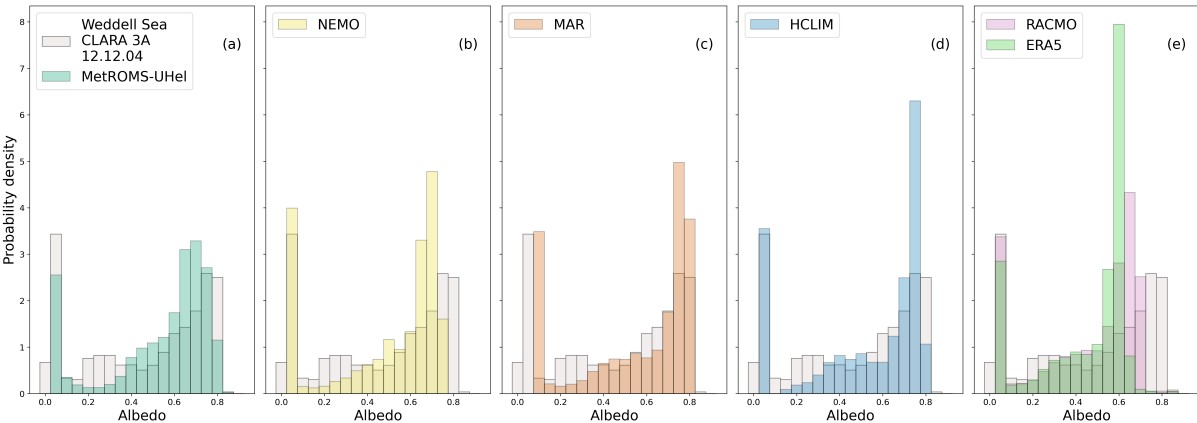

**Figure 12.** Panels a-e: Sea ice albedo distribution over a 2 799 169 km$^2$ Weddell Sea area (the area is defined by the smallest, HCLIM domain size). .

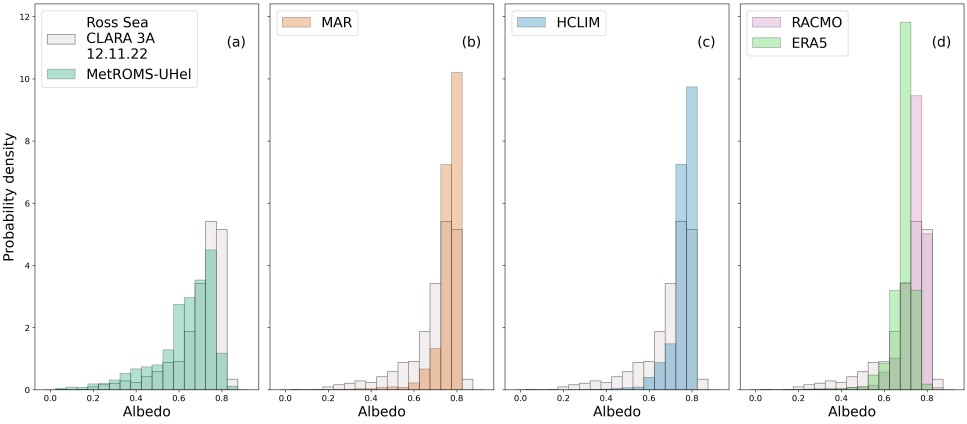

**Figure 13.** Sea ice albedo distribution of sea ice albedo over a 3 200 661 km$^2$ Ross Sea area (the area is defined by the smallest, HCLIM domain size). NEMO model runs are available only for the ISPOL case and not for the Marsden field campaign. .

## 5 Discussion

As described in Sect. 3, each of the used models in this study is different in how they handle sea ice and surface albedo. We
tested the models against various observational data – from detailed in-situ data from the ISPOL and Marsden field campaigns, to small-area Landsat 9 and Sentinel 2 albedo products, to large-scale CLARA-3A products. ISPOL campaign took place in December 2004 on a single ice floe in the Weddell Sea, where RV Polarstern was moored. The weather was warm, with air temperature mostly above -5°C and even around zero degrees during the first week of December. The Marsden campaign took place in November 2022, over the land-fast sea ice of McMurdo Sound. Located a few kilometres north of Ross Island, the





field campaign took place in a challenging region for models. Dry and cold conditions held over most of the campaign, with a steady temperature increase from -25°C to -3°C, and a snowstorm in the last week of the campaign.

The in-situ measurements are below what models can resolve, and the surface albedo can be driven by localised effects, such as topography or blowing snow. When compared to measurements from the ISPOL and Marsden campaign sites, the models reproduced the conditions observed during the ISPOL campaign but did not accurately simulate the conditions during

the Marsden campaign. The ice floe, to which RV Polarstern was moored during the ISPOL campaign, had up to a 1 m snow layer on top. In December 2004, observations showed the sea ice had a mean albedo of 0.78 (with a standard deviation of 0.06 due to temporal variability), and 0.82 (0.05) in two measurement stations on the ISPOL ice floe. The diurnal variability observed at the ISPOL campaign is driven by near-melt snow conditions during daytime, refreezing during nighttime, and solar zenith angle. All modelled albedos, except RACMO and ERA5, fell within the observed ranges. The MetROMS-UHel model

accurately captured both the magnitude and diurnal variability of snow albedo for the ISPOL case. In contrast, RACMO's prescribed albedo was consistently about 0.1 too low throughout the ISPOL campaign season.

The snow and ice conditions in the Marsden campaign area were different, with land-fast ice ranging from 1 to 4 meters thick and covered by a very thin, few to few tens of cm thick snow layer. The observed mean surface albedo was 0.79, largely influenced by variations in drifting snow accumulation patterns over the continuous, but thin, snow cover. All models, except

RACMO, simulated sea ice albedo as being dominated by cold, snowy conditions and were unable to account for the patchiness and drifting snow patterns observed in the region. RACMO accurately predicted the observations from the Marsden campaign, despite relying on Ebert and Curry (1993) monthly averages of sea ice albedo for its predictions. The Marsden campaign site is located a few kilometres north of Ross Island. The MAR model and ERA5, both of which have the lowest spatial resolutions at 25, and ∼28 km respectively, inaccurately represented the Marsden site as being on land rather than on sea ice.

The Marsden campaign also recorded albedo measurements from a drone at a height from 30 to 70 meters, capturing the averaged albedo in 0.02-0.04 km² areas over thick sea ice (∼2.3 meters) and thinner sea ice (∼1.2 meters). While the thicker ice was covered by a heterogeneous snow layer ranging from a few centimetres to a few tens of centimetres thick, with small patches of bare ice, the thinner ice was mostly bare with small patches of snow a few centimetres thick. The oceanic models NEMO and MetROMS-UHel, coupled with their respective sea ice models, categorise sea ice by thickness, enabling

comparisons between similar ice types and their corresponding surface albedos. NEMO's snow-free conditions allowed us to better understand the modelled bare ice characteristics. NEMO's bare ice albedo depends mainly on ice thickness, with a maximum albedo value of 0.5 for 1 m thick sea ice, and approaching an albedo value of 0.6 for 1.5 m and thicker ice. This is an overcast albedo value from which the clear-sky value is derived using cloud fraction. In the MetROMS-UHel model, the albedos for both ice categories are snow-dominated. While this model successfully reproduced the average albedo measured

by the drone over thicker ice, it did not accurately capture the albedo over thinner ice. Over thinner ice, the measurements were dominated by bare ice with a patchy snow cover. Both NEMO and MetROMS-UHel account for fractional snow coverage when calculating albedo, but they simulated entirely snow-free or fully snowy conditions, respectively.

Models continue to struggle with accurately reproducing snow albedo variability, particularly under near-melt conditions and conditions when snow cover is very thin or patchy. However, since the primary application of these regional atmospheric





and oceanic models is over larger geographic areas, broader-scale albedo comparisons are essential to determine whether the models align with observations. We further tested the models on two different scales. First, we used high-resolution albedo images from Landsat 9 and Sentinel 2 over approximately $260 \times 260$ km$^2$ and $110 \times 110$ km$^2$ area, respectively, at the Marsden campaign site. Second, we employed 25-km resolution CLARA-A3 albedo products to examine the entire Weddell and Ross seas during the respective ISPOL and Marsden campaigns.

High-resolution satellite images revealed a wide range of albedo values across the Marsden campaign site, from around 0.1 for open ocean to 0.75 or higher for land-fast ice, depending on snow cover, with various values in between. Albedo variability was evident at km-scales, yet the models failed to capture these details. Instead, variations in sea ice concentration and regional snow cover differences led to a much narrower (0.4-0.7) distribution of albedos in the models. At the broader scale of the Weddell and Ross seas, the surface albedo is primarily determined by sea ice concentration fields, with HCLIM, MAR,
and RACMO all using ERA5 input for this purpose. In contrast, MetROMS-UHel and NEMO model the sea ice evolution, leading to differences in both sea ice concentration and albedo. Nonetheless, the snow and ice albedo parameterisation remains equally important. RACMO and ERA5 generated surface albedos that were 0.25 lower than CLARA-3A satellite products and other models over the Weddell Sea. Over the Ross Sea during the Marsden campaign, RACMO and ERA5 were more in line with the CLARA-A3 albedos. The modelled albedo in both RACMO and ERA5 is based on monthly albedo values from
Ebert and Curry (1993). This approach means that while the predicted albedo is sometimes accurate, it often is not, as the actual synoptical evolution is not taken into account. Several authors have highlighted the challenges ERA5 faces in accurately representing sea ice albedo over the Arctic (Pohl et al., 2020; Müller et al., 2024; Batrak et al., 2024), and our findings confirm that these challenges exist for the Antarctic as well.

  Overall, the in-situ and drone-based albedo measurements collected during the ISPOL and Marsden campaigns offer a
valuable resource for model evaluation and calibration, particularly in capturing temporal changes. However, we recognise that evaluating models based on single-point observations is challenging, as these observations may not accurately reflect the average situation for the larger grid cells used in climate models. While validations against satellite observations provide a useful complement by revealing spatial variability in albedo, their temporal resolution is already only daily at maximum and limited to cloud-free conditions. As the ocean around Antarctica is one of the regions with the most persistent cloud cover, this
reduces considerably the number of images that observe sea ice.

## 6 Conclusions

This model comparison study examined the sea ice representation in three regional atmospheric models (HCLIM, MAR, RACMO), ERA5 reanalysis, and two regional ocean models (MetROMS-UHel, NEMO). We concentrated on sea ice albedo. The regional atmospheric and ocean models rely on sea ice albedo parameterisations. However, due to the scarcity of obser-
vational data, especially over Antarctica, and the limitations imposed by model resolution and computational costs, simplified snow and ice albedo parameterisations are applied. The level of accuracy representing the ice and snow albedo required depends on the specific use case of a particular model, but even small adjustments in snow and ice albedo parameterisations



can lead to significant improvements in reproducing the observed sea ice albedo, as well as other variables, such as surface temperatures. The key takeaways from our study are:

1. Accurate parameterisation of bare ice and snow albedo is essential for model performance. Fine-tuning these parameters can enhance the model's ability to reproduce observed conditions. For example, initial tests with the HCLIM model revealed a significant warm bias in surface temperatures, with discrepancies as large as 5°C over sea ice compared to measurements from the ISPOL campaign. This bias was attributed to snow grain size distribution unsuitable for Polar regions, which also led to too low modelled albedo. Updating the snow grain size distribution led to albedo levels similar

to observations and removed the warm bias. Furthermore, ERA5 (and RACMO) both use time-interpolated monthly albedo values based on Ebert and Curry (1993), which has been identified as a limitation in this and several other studies (Pohl et al., 2020; Müller et al., 2024; Batrak et al., 2024).

  2. In drier sea-ice regions like the Ross Sea, the key issue affecting the performance of albedo models is the treatment of fractional snow cover. When models simulate a fully snow-covered surface, their albedo values tend to be significantly

higher than the observed albedo values, which incorporate substantial areas of snow-free sea ice.

  3. Sea surface albedo can be influenced by highly localised factors such as small-scale variations in sea ice concentration, and patchy or blowing snow. As a result, higher-resolution models do not necessarily outperform lower-resolution models if these localised effects are not accounted for. For instance, although HCLIM operates at a 2.5 km spatial resolution, its sea ice concentration fields are derived from the 0.25° (∼28 km) resolution ERA5 data and it employs a simple

one-dimensional thermodynamic sea ice parameterisation scheme, which limits its ability to capture these finer-scale processes.

  4. Comparing the modelled total (ice, snow and ocean) grid-box averaged sea ice albedo to in-situ measurements near the margins of the sea ice extent (with lower sea ice concentrations) or on land-fast sea ice (with highest sea ice concentrations) is problematic. In this study, HCLIM, MAR, and RACMO all used ERA5 sea ice concentration fields, but

MetROMS-UHel and NEMO calculate these fields prognostically. For instance, the ISPOL campaign site was at the very margins of NEMO's sea ice extent. Furthermore, the location of the Marsden campaign site was complicated to the models – ERA5 and MAR consider the Marsden CS to be on land, not on sea ice. When evaluating ERA5 and MAR albedo during the Marsden campaign, the values from the closest sea-ice gridpoint were used. Since HCLIM, MAR, and RACMO use sea ice concentrations from ERA5, it implies that these models extrapolate the sea ice concentrations to

areas classified as land in ERA5.

Surface albedo over Polar sea ice is complex, and this study focused only on first-order effects, excluding factors such as the cloud and solar zenith angle dependence of surface albedo. In study cases characterised by significant variation in surface types, such as during the spring/summer season, it is primarily uncertainties in the parameterisation of these surface types that influence the modelled surface albedo, rather than cloud effects (Jäkel et al., 2023). While future research should include a



more comprehensive evaluation of cloud impacts, as explored by Jäkel et al. (2023) and Foth et al. (2023), there is still room for improvement by refining the aspects of albedo discussed in this paper.

Coupling advanced radiative transfer models with regional climate or ocean models, as demonstrated in RACMO (though limited to land, not sea ice), MetROMS-UHel, and SNICAR-ADv4 (Whicker et al., 2022), represents a major step forward in simulating surface processes. This approach enables a detailed representation of the optical properties of both snow and ice, 615 thereby enhancing the accuracy and performance of coupled model systems.

*Data availability.* ERA5 data are available via the Copernicus Climate Data Store (Hersbach et al., 2020). The used satellite images, and model data are archived at Zenodo (Verro, 2025). The ISPOL meteorological data is provided by König-Langlo (2005), albedo data by Nicolaus et al. (2009a), and first-year ice (FYI) snow and ice thickness measurements by Nicolaus et al. (2009b). The snow thickness measurements on second-year ice (SYI) conducted by FMI are available at Verro (2025). Marsden campaign meteorological data and snow 620 and ice thicknesses are being prepared to be published (Dadic et al. in prep.). The albedo measurements are available at Verro (2025), and a data release by Pirazzini et al. is currently in preparation.



## Appendix A: Meteorological comparison extended

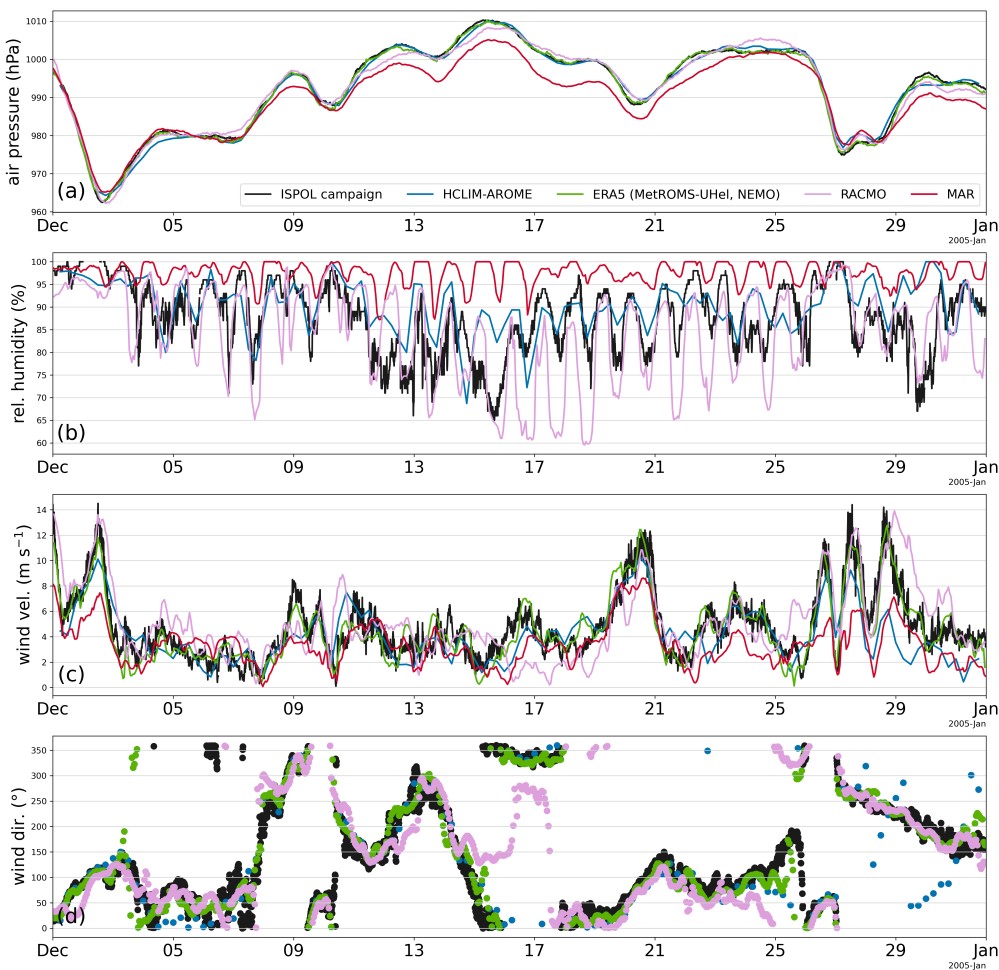

**Figure A1.** Weather conditions during the ISPOL experiment measured at the RV Polarstern (black) compared with the model output from HCLIM (blue), RACMO (pink), MAR (orange) and ERA5 reanalysis (green). Panel a: air pressure, b: relative humidity, c and d: wind velocity and direction.





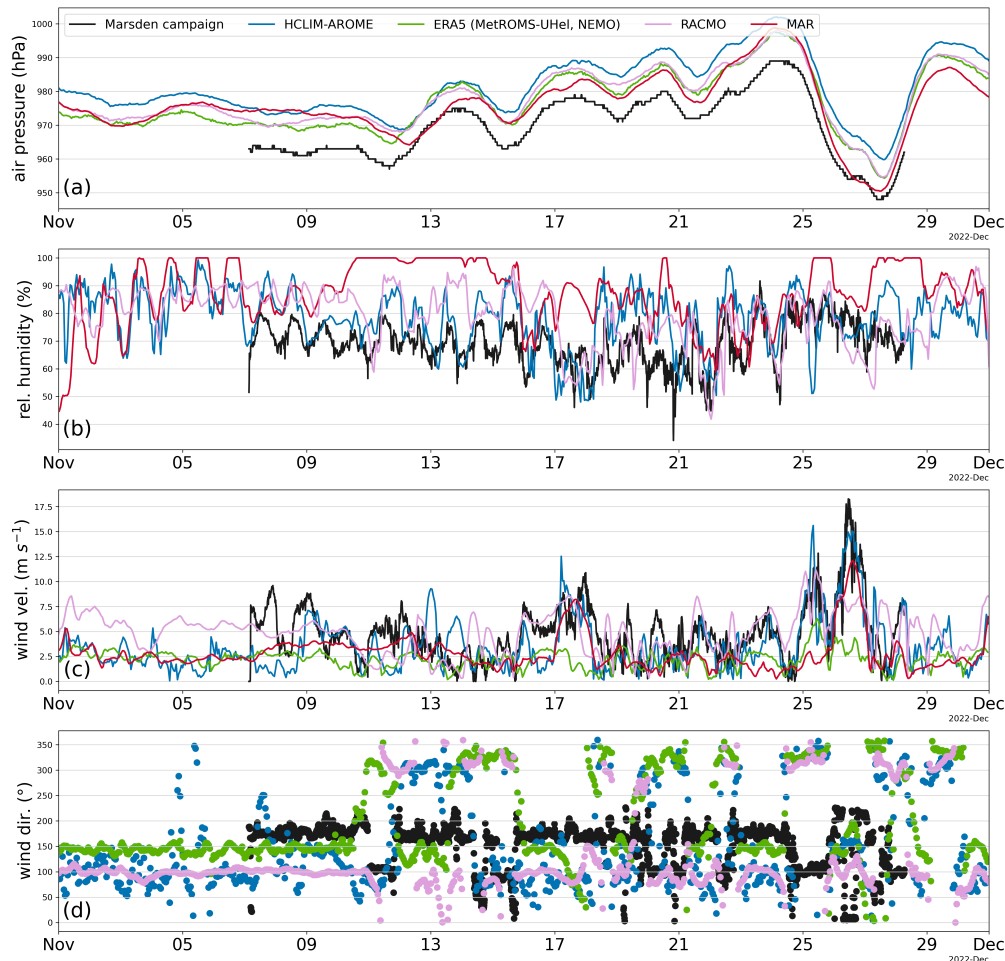

**Figure A2.** Weather conditions during the Marsden campaign compared with the model output from HCLIM (blue), RACMO (pink), MAR (orange) and ERA5 reanalysis (green). Panel a: air pressure, b: relative humidity, c and d: wind velocity and direction.



**Appendix B: Sea ice albedo distributions over Weddell and Ross Seas, modelled snow and ice only.**

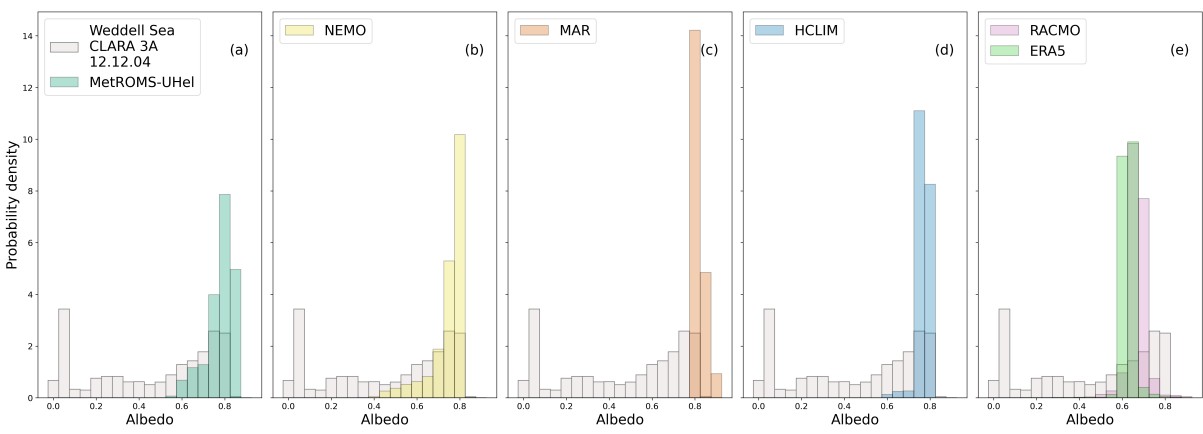

**Figure B1.** Probability distributions of sea ice albedo over a 2 799 169 km$^2$ Weddell Sea area (the area is defined by the smallest, HCLIM domain size). For the model output on panels (a)-(e), only the contributions from sea ice and snow are shown, with ocean albedo removed.

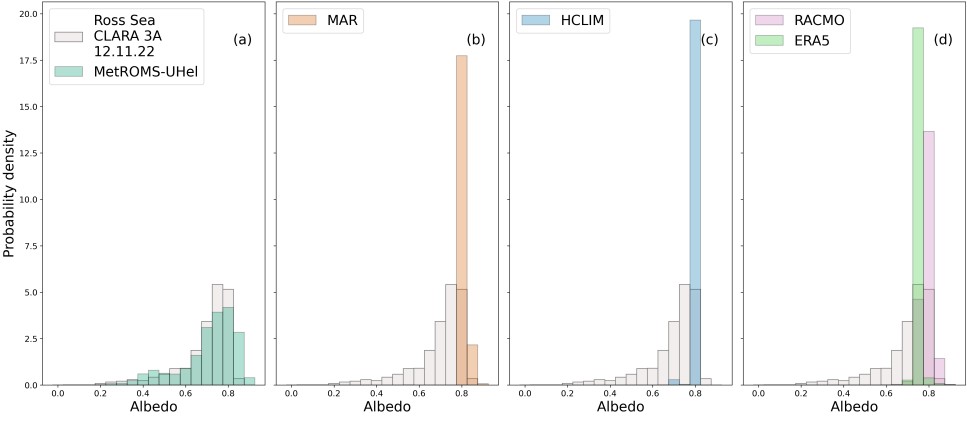

**Figure B2.** Probability distributions of sea ice albedo over a 3 200 661 km$^2$ Ross Sea area (defined by HCLIM domain size). For the model output on panels (a)-(d), only the contributions from sea ice and snow are shown, with ocean albedo removed. NEMO model runs are available only for the ISPOL case and not for the Marsden field campaign.

*Author contributions.* KV prepared the manuscript with contributions from all co-authors. CÄ and PU provided model output and descriptions for MetROMS-UHel. DM and XF provided the same for MAR and NEMO; KV and WJvdB for HCLIM; CvD and WJvdB for RACMO. Satellite products were provided by GT and BDM. RP and RD contributed with Marsden campaign data and description. RP and MJ with ISPOL data.




*Competing interests.* At least one of the (co-)authors is a member of the editorial board of The Cryosphere.

*Acknowledgements.* KV, CÄ, RP, DM, WJvdB, PU, CvD, and XF acknowledge support from the PolarRES project, which has received

funding from the European Union's Horizon 2020 research and innovation programme call H2020-LC-CLA-2018-2019-2020 under grant agreement number 101003590. The McMurdo field campaign was funded through the New Zealand Marsden Fund project 21-VUW-103, with support from Antarctica New Zealand. RP and RD acknowledge Julia Martin for Marsden field data collection; Brian Anderson, Martin Schneebeli, Matthias Jaggi, Inga Smith, Huw Horgan, Greg Leonard, Natalie Robinson, Oliver Wigmore, Wolfgang Rack, and Darcy Mendeno for providing field equipment. CÄ and PU wish to acknowledge CSC – IT Center for Science, Finland, for computational resources.

KV acknowledges the assistance of ChatGPT, which was only used for language editing purposes.



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
