# Peer review of "How well do the regional atmospheric and oceanic models describe the Antarctic sea ice albedo?"

_EGUsphere, 2025_

## Referee Comment (RC1)

Referee comment on "How well do the regional atmospheric and oceanic models describe the Antarctic sea ice albedo?" by Verro et al.

The study investigates the model performance of several regional oceanic and regional atmospheric models with respect to the representation of sea ice properties (surface albedo), snow and ice thickness in Antarctica. The authors compare the model results with measurements from ground-based, drone-based, and satellite-based observations. Publications of this kind, which make use of so many models, are rare. I appreciate the work the authors have done to bring together observational data and modelling results.

Overall, the authors have done a good job of providing an overview of the models used and the observational dataset. The intercomparison of modelled surface albedos is presented well. Some suggestions for improving the content are provided below. I can recommend the publication after minor revisions.

General Comments

1. The abstract is a bit long. It goes into a lot of detail that is not necessary at this point. I would like to see more quantitative results in the abstract instead.
2. Section 5 reads like a summary. There is no further discussion of the results, which were very nicely presented in the previous section. I recommend deleting the whole section.

Minor/Specific Comments

1. p1l2, p2l41, p3l76: "regional climate model" → regional atmospheric climate model
2. p1l17: "RACMO and ERA5 predict significantly darker sea ice over the Weddell Sea during the…": I would omit "darker" here. A surface can appear darker under cloudy conditions even though its albedo is the same or even higher.
3. p2l50: "sea ice albedo parametrisation recommendations, such as those given in Ebert and Curry (1993);" Ebert and Curry give basically climatological-based values. Maybe emphasize that.
4. p2l52: "surface temperature": Is it the skin temperature?
5. p4l107: "Broadband albedo was measured from pyranometers installed": give type of pyranometers similar to section 2.2
6. p4l108: "FIMR station": What means FIMR?
7. p4l113: "The accuracy is approximately 3% for the shortwave radiation measurements (Vihma et al., 2009)." Is this the uncertainty of the irradiance or the derived albedo?
8. p5 Figure caption: "The image shows an overlay of Landsat surface temperatures over a Landsat grayscale visible image on October 10, 2022." – It is not discussed in the main text. Use larger fonts in Fig. 1b and lon/lat grid as in Fig. 1a to get a better orientation.
9. P6l139: "… but the broadband albedo products can be calculated at 30 m and 20 m resolutions": Give retrieval uncertainty here. Currently, the numbers are mentioned in Sec. 4.5.
10. P8l169: "The shortwave radiation in the atmosphere and the coupled ice/snow layer is handled by a Delta-Eddington multiple scattering radiative transfer model (Briegleb et al., 2007)." Does the model consider clouds?
11. p10 Figure 2: Please use larger fonts. Think about to show a distribution of the albedo difference in addition.
12. P11l274: "The albedo is derived by using separate values for near-infrared and visible light." How is the broadband albedo derived from the albedo of the two spectral regions?
13. P12l283: "ERA5 considers the Marsden CS to be on land, not on sea ice." What are possible effects?

14. P12l294: "During the ISPOL campaign, the weather was warm for this location, with the air temperature mostly above -5°C and even around zero degrees during the first week of December." Already mentioned before. Can be removed.
15. P12l295: "HCLIM reproduces the surface temperature well" Maybe use "best" instead of "well"?
16. P18l392: "However, NEMO output for the Ross Sea in November 2022 was unavailable, as the model data extends only up to 2018. Therefore, data from November 2004 for the same region is used instead." – Does this mean that the distributions at Marsden in 2022 are being compared with those in 2004? Why was 2004 chosen? What makes this year a representative sample of 2022?
17. P18l407: "The spread of the drone-based albedo probability distributions, which represent the measurement uncertainty during ~10 minute flight, …" – Why does the distribution represent the measurement uncertainty? Rather, it should reflect the variability of the surface.
18. P18l412: "Hence, we can argue that it also represents the spatial albedo variability, though biased toward the albedo of the most frequent surface type that happened to occur right below the pyranometer." Can we really say here that temporal variability can be taken as a proxy for spatial variability? Albedo variability also depends on atmospheric parameters such as SZA and cloud cover, which are certainly reflected in the temporal variability within the one-month period. However, for a 10-minute flight, I would assume that these parameters have less effect.
19. P18l420: "The discrepancy between the observations and themodels over the thinner ice is large: Δmean = 0.2, and 0.14 for MetROMSUHel and NEMO respectively." Is the comparison meaningful, as different years with probably different conditions are taken into account?
20. P19 Section 4.5: It is useful to show the spatial variability of satellite and model data. However, the authors could also use high-resolution satellite data to compare albedo directly with ground-based observations.
21. P19L439: "Landsat 9 albedo observations on the 1st of November in Fig. 7, and Sentinel 2 observations on the 14th of November, 2022 … " - Is this an example that can be used to represent the whole period?
22. P20l444: "However, the albedo over land is about 0.06 higher in the Sentinel 2 image compared to the Landsat 9 image." – The manuscript is about sea ice. Therefore, I would limit the discussion to that.
23. P21l482: "This peak does not come from sea ice concentrations in the area but from the sea ice albedo parameterisation." – Can you elaborate this statement?
24. P22l489: "The spatial distribution and density distribution of the observed albedo are best reproduced by the MetROMS-UHel model." – This statement suggests that CLARA-A3 is the truth. How large is the retrieval uncertainty of the CLARA-A3 product? Perhaps it is better to say here that the MetROMS-UHel model shows the best agreement with the satellite product.
25. P24 Section Discussion: This section is more of a summary than a discussion. Apart from some text at the end that could be moved to the 'Conclusions' section, I don't see much new information here.
26. P26 Section Conclusion: It would be good to support the conclusions with some numbers to make them more quantitative.

Technical Comments

1. check format of citations for example:

p2l39: "0.06.(Warren,", p2l48: "by (Debernard et al., 2017…", p3l54: "as in ERA5 Hersbach et al. (2020)", p4l107: "snow. (Hellmer et al., 2006).", p6l148: "zenith angles Traversa and Fugazza (2021)", p8l167: "scheme Lipscomb and Hunke (2004)"

2. p7 Table 1: first line "absorption/scattering" check hyphen separation, last line "othewise" typo

3. p8l194: "Melt pond properties as given by the physical level-ice scheme characterised by…" → are characterised

4. P8l182: "The model runs at 0.25° resolution" → 1/4° as used in Table 1

5. p9l203: "The regional atmospheric model HARMONIE Climate (HCLIM, Belušic et al. (2020)) cycle 43 using the non-hydrostatic …" → model HARMONIE Climate cycle 43 (HCLIM, Belušic et al., 2020)

6. p10 Figure 2: Please use larger fonts. Think about to show a distribution of the albedo difference in addition.

7. P11l269: "at the lateral boundaries (van Dalum et al. submitted to the Cryosphere)." Cite the discussion paper.

8. P12 Figure 3: Think about to move the legend from Fig 3c to the top of the figure.

9. P34l691: "https://doi.org/https://doi.org/10.1029/2023EA003482": remove first "https://doi.org/", there are several more references with similar issues

10. P34l700: please update reference

11. P35l738: please update reference

12. P38l841: please update reference

---

## Author Comment (AC1)

**Response to the RC1 #1 comment on "How well do the regional atmospheric and oceanic models describe the Antarctic sea ice albedo?" by Verro et al.**

We thank the referee for the recommendation to accept the publication after minor revisions and the suggested changes. We hereby address most of the comments, questions and suggestions.

1) The abstract is a bit long. It goes into a lot of detail that is not necessary at this point. I would like to see more quantitative results in the abstract instead.

   We have edited the Abstract to have some quantitative results, and shortened it somewhat (from 462 words to 421):

   "A realistic representation of the Antarctic sea ice surface albedo, especially during the spring and summer periods, is essential to obtain reliable atmospheric and oceanic model predictions. We used regional climate (HCLIM, MAR, RACMO), regional oceanic (MetROMS-UHel, NEMO) models and ERA5 reanalysis to investigate how well these models describe the basic sea ice characteristics: sea ice albedo, snow and ice thickness. We analyse models against a range of observations, including in-situ measurements from the ISPOL (Weddell Sea, Dec. 2004) and Marsden (McMurdo Sound, Nov. 2022) field campaigns, as well as drone and satellite data.
   Models perform well in reproducing the sea ice in certain conditions: during the ISPOL campaign, characterised by thicker snow cover and mild weather that resulted in daytime melt-driven albedo changes and nighttime refreezing in the snow-covered sea ice most models did well; MetROMS-UHel (average albedo = 0.77±0.09 ), NEMO (0.81± 0.02), HCLIM (0.78± 0.03) and MAR (0.83±0.01) reproduce mean values found in observations (0.78±0.06 and 0.82±0.05), and MetROMS-UHel captures even the observed diurnal albedo variability.
   However, all models had difficulty reproducing the sea ice conditions in the McMurdo Sound. MetROMS-UHel (average albedo = 0.85±0.00 ) and HCLIM (0.84± 0.00) showed high sea ice albedo without any temporal variability, while MAR (0.74±0.13), had a significant increase in surface albedo, from 0.55 to 0.85, when a thin layer of snow fell within the previously bare ice grid. None of these model behaviors matched the Marsden field campaign observations (0.79±0.03). Over the colder and drier sea-ice regions with thinner or patchy snow cover, the key issues affecting the accuracy of albedo models are the treatment of fractional snow cover and the snow albedo dependence on snow depth. Over the broader Weddell and Ross seas, sea ice albedo is primarily determined by sea ice concentration fields, but albedo parameterisations are still relevant: RACMO and ERA5 predict significantly lower albedo sea ice over the Weddell Sea during the ISPOL campaign.

The complexity of sea ice albedo parameterization significantly influences model performance. ERA5 and RACMO use fixed albedo values based on \citet{ebert1993}. HCLIM incorporates an intermediate-complexity snow model, where snow albedo depends on snow density and bare ice albedo on temperature. In contrast, more advanced models like MetROMS-UHel apply radiative transfer schemes that calculate albedo from the inherent optical properties of the surface. Integrating such sophisticated schemes into regional climate or ocean models represents a major step forward in accurately simulating surface energy processes."

2) Section 5 reads like a summary. There is no further discussion of the results, which were very nicely presented in the previous section. I recommend deleting the whole section.

To improve the compactness of the manuscript, we are reviewing Section 5 for possible shortening or, even, deletion.

1) p1l2, p2l41, p3l76: "regional climate model" -> regional atmospheric climate model

We have changed the wording.

2) p1l17: "RACMO and ERA5 predict significantly darker sea ice over the Weddell Sea during the…": I would omit "darker" here. A surface can appear darker under cloudy conditions even though its albedo is the same or even higher.

Thank you for the remark. We agree that surface albedo can be modified by cloud cover, but in this case, we have shown that the problem of sea ice albedo in RACMO and ERA5 is more than that. Firstly, Fig. 4 shows that both models are consistently "darker" throughout December 2004 over the ISPOL campaign region.  The sea ice albedo in both models is again consistently darker over the whole sea ice in the Weddell Sea also on Fig. 10.  It is a problem of parametrization.

We have changed the wording to: "Albedo parameterisations are still relevant: RACMO and ERA5 predict significantly **lower** albedo sea ice over the Weddell Sea during the ISPOL campaign, while their predictions align better with observations over the Ross Sea during the Marsden campaign."

3) p2l50: "sea ice albedo parametrisation recommendations, such as those given in Ebert and Curry (1993);" Ebert and Curry give basically climatological-based values. Maybe emphasize that.

We agree that it is important to emphasise the usage of climatological-based values, so we added a sentence to the same section: "In the simplest cases, sea ice albedo is

presented as climatological mean values, as in ERA5 (Hersbach et al., 2020) and RACMO which are using Ebert and Curry (1993) parameterisations." See also our answer to point 12.

4) p2l52: "surface temperature": Is it the skin temperature?

Skin temperature would be a more accurate word to use, so we have changed it: "Sea ice albedo is typically parameterised as a function of one or more variables, including air temperature, skin temperature, snow/ice type, snow grain size, snow depth/density, sea ice thickness, cloud cover fraction and solar zenith angle."

5) p4l107: "Broadband albedo was measured from pyranometers installed": give type of pyranometers similar to section 2.2

We have included a sentence: "Broadband albedo was measured from a pair of Kipp & Zonen CM5 (FIMR station) and Kipp & Zonen CM22 (AWI station) pyranometers, installed at about 1.5 m above the surface.."

6) p4l108: "FIMR station": What means FIMR?

We have included a sentence that introduces the acronyms:
"The Alfred Wegener Institute (AWI) measurement station was operated over the FYI, and the Finnish Institute of Marine Research (FIMR) station obtained readings from the SYI."

7) p4l113: "The accuracy is approximately 3% for the shortwave radiation measurements (Vihma et. al., 2009)." Is this the uncertainty of the irradiance or the derived albedo?

It is not albedo uncertainty, but a systematic radiation measurement error derived from comparisons between AWI and FIMR independent measurements.

8) p5 Figure caption: "The image shows an overlay of Landsat surface temperatures over a Landsat grayscale visible image on October 10, 2022." – It is not discussed in the main text. Use larger fonts in Fig. 1b and lon/lat grid as in Fig. 1a to get a better orientation.

We have added sentence to p4l95:
"Figure 1a shows the locations of these campaigns, with a zoom-in in Figure 1b on the Marsden campaign site, which was located in a more topographically complex region."

And: p5l126:
"Figure 1b also shows Landsat surface temperatures, illustrating potential sources of different sea ice characteristics in the region."

We have increased the font sizes in Fig. 1 and added the phrase '...of the Marsden field campaign in Polar Stereographic coordinates' to the caption of panel 1b. However, we believe that the orientation indicated by the compass (which we made larger) is sufficient.

[Figure]

9) P6l139: "… but the broadband albedo products can be calculated at 30 m and 20 m resolutions": Give retrieval uncertainty here. Currently, the numbers are mentioned in Sec. 4.5.

We have moved the sentence for Sec. 4.5 to p6l150:
"The uncertainty of the Landsat 9 albedo product, based on the applied methodology described in Sect. 2.3, is ±0.02 in polar regions (Traversa et al., 2021). For Sentinel 2 albedo imagery, an uncertainty of ± 0.05 was estimated by Naegeli et al. (2017); Di Mauro et al. (2024)"

And, included a shorter sentence to p19l483:
"The uncertainty of the Landsat 9 albedo product is ±0.02 in polar regions, while that of the Sentinel-2 albedo imagery is ±0.05."

10) P8l169: "The shortwave radiation in the atmosphere and the coupled ice/snow layer is handled by a Delta-Eddington multiple scattering radiative transfer model (Briegleb et al., 2007)." Does the model consider clouds?

Yes, the shortwave radiation given to the Delta-Eddington model is calculated using cloud cover . We have added this information to Table 1 and to the Sect. 3.1: "The shortwave radiation forcing given to the Delta-Eddington model is calculated using total cloud cover, humidity and solar zenith angle."

11) p10 Figure 2: Please use larger fonts. Think about to show a distribution of the albedo difference in addition

We have included histograms © of sea ice albedos of the default and updated HCLIM simulations shown on the map on (b). We have also made the fonts larger.

[Figure]

12) P11l274: "The albedo is derived by using separate values for near-infrared and visible light." How is the broadband albedo derived from the albedo of the two spectral regions?

We have expanded the description of RACMO sea ice albedo, to answer Referee's questions 12 and 3.

P11l280: "The albedo is based on Ebert and Curry (1993) and thus does not depend on snow conditions. Instead, monthly sea ice albedo values for the Arctic Ocean, valid on the 15th of each month, are linearly interpolated to the forecast time. For the Antarctic, the seasonal cycle is shifted by six months. The bare sea ice albedo represents summer months, while the dry snow albedo is applied during winter months, both with direct and diffuse components. The albedo is derived by
using separate values for four spectral bands in the near-infrared and visible light, and the broadband albedo is the sum of the spectral albedos weighted by the relative solar flux in each wavelength region, and further weighted by the direct and diffuse fractions. While this parametrisation scheme intends to capture key surface, cloud, and atmosphere effects, the albedo values are climatological means, and do not depend on the state of the surface and atmosphere within RACMO model. For example, the effect of fresh snowfall on the albedo on sea ice during summer is therefore often not captured properly, as the albedo model does not take the actual conditions into account. "

13) P12l283: "ERA5 considers the Marsden CS to be on land, not on sea ice." What are possible effects?
To analyse the sea ice properties, instead of land, the values from the closest sea-ice gridpoint were used. This was done in the case of ERA5, but also for MAR.

The Marsden campaign area is highly complex; the proximity of Ross Island and the Ross Ice Shelf results in rapidly changing sea ice properties over short distances. This spatial variability was also reported by field observers and is evident in Table 3 and Figure 1b, where both snow/ice thickness and surface temperature differ significantly between sites. In both cases, the nearest sea ice model grid point is not located at the

CS site but towards NIS or TRS sites or farther. Snow and sea ice thicknesses were lower at NIS (Table 3), with snow cover described as thin and patchy rather than uniformly thick as in CS. As a result, surface albedo was more strongly influenced by bare ice, as also seen in drone observations (Figure 6).

None of the models could reproduce this level of detail. ERA5, relying on climatological averages, distinguishes only by sea ice concentration. ERA5 and RACMO use the same sea ice model, but with improved resolution, RACMO models the CS on sea ice. We would expect ERA5 to look more like RACMO albedo on Fig. 4. However, it is important to note that RACMO (as well as HCLIM and MAR) use ERA5 sea ice concentrations, and each model interpolates and extrapolates the concentrations to their own grid.

Additionally, due to the region's complexity, even atmospheric variables are poorly represented by the models (Figure A2). While observations taken farther from land—such as during the ISPOL campaign—would offer a cleaner comparison, we work with the data available.

We have mentioned some of the difficulties in in Sect. 4.1, 4.3 and in the Conclusions, but we have now also added to Sect 4.2:

"Comparisons with the Marsen CS measurements come with a caveat: to analyse the sea ice properties, instead of land, the EAR5 and MAR model values from the closest sea-ice gridpoint were used. In both cases, the nearest sea ice model grid point is not located at the CS site but towards NIS or TRS sites or farther. The proximity of Ross Island and the Ross Ice Shelf results in rapidly changing sea ice properties over short distances, as described in Sect. 4.3 and 4.4. While observations taken farther from land — such as during the ISPOL campaign — offer a cleaner comparison, we work with the data available."

14) P12l294: "During the ISPOL campaign, the weather was warm for this location, with the air temperature mostly above -5°C and even around zero degrees during the first week of December." Already mentioned before. Can be removed.
See next answer.

15) P12l295: "HCLIM reproduces the surface temperature well" Maybe use "best" instead of "well"?
Removed repetitive sentence, and edited the next sentence: "HCLIM reproduces the surface temperature observed during the ISPOL campaign best (with a mean difference of 0.2∘C),
as well as the surface pressure, wind speed and direction."

16) P18l392: "However, NEMO output for the Ross Sea in November 2022 was unavailable, as the model data extends only up to 2018. Therefore, data from November 2004 for the

same region is used instead." – Does this mean that the distributions at Marsden in 2022 are being compared with those in 2004? Why was 2004 chosen? What makes this year a representative sample of 2022?

Using 2004 data to simulate 2022 conditions would only be valid if ice and snow conditions were highly uniform across the area and remained stable from year to year. 2004 was selected, as we have NEMO output for 2004 for the ISPOL campaign analysis. The model's intrinsic bare ice characteristics per category remain consistent year to year, while the presence of ice types, the fractional snow cover, and the cloud coverage change. Therefore, we cannot guarantee that 2004 NEMO output is representative for 2022.

In the revised manuscript, we will remove NEMO from the direct comparison with the 2022 Mardsen data on Fig. 6. Instead, we include the 2004 NEMO results to illustrate the model's capabilities, comparing them with MetROMS-UHel output for 2004.

Comparison of the observed sea ice albedo with the modelled sea ice albedo of MetROMS-UHel for November 2022:

[Figure]

Comparison of the modelled sea ice albedo of NEMO with the modelled sea ice albedo of MetROMS-UHel for November 2004:

[Figure]

We have edited the text:

"However, NEMO output for the Ross Sea in November 2022 was unavailable, as the model data extends only up to 2018. Therefore, we compared MetROMS-UHel with the Marsden campaign measurements from November 2022, and with NEMO for November 2004."

Furthermore:

"The discrepancy between the observations and the models over the thinner ice is large: ∆mean = 0.2 for MetROMS-UHel. In this case, the MetROMS-UHel modelled albedo is snow-dominated, as both ice categories are covered with a layer of snow. The same situation occurs in 2004, where the MetROMS-UHel albedo is also dominated by snow, but NEMO had snow-free conditions. NEMO's bare ice albedo depends mainly on ice thickness, with a maximum albedo value of 0.5 for 1 m thick sea ice, and approaching an albedo value of 0.6 for 1.5 m and thicker ice, with additional adjustments based on cloud fraction."

17) P18l407: "The spread of the drone-based albedo probability distributions, which represent the measurement uncertainty during ~10 minute flight, …" – Why does the distribution represent the measurement uncertainty? Rather, it should reflect the variability of the surface.

During a single flight, the drone measured the vertical profile of albedo over the same coordinate point, increasing the footprint of the downward facing pyranometer while increasing the height above the surface. Starting from the altitude of 30 m and above, the measured albedo did not change with altitude, meaning that the spatial distribution of surface albedo variability included in the footprint of the downward facing pyranometer did not change with increasing footprint radius. Hence, the small variations in the drone-based albedo observed from 30 m above the surface and at higher altitudes are

associated to measurement errors (caused for instance by small vibrations of the drone, and small deviations from the horizontal alignment of the pyranometers when compensating for changes in wind speed) rather than to spatial variability. We edited the text to better explain this point, including:

"Starting from the altitude of 30 m and above, the measured albedo did not change with altitude, meaning that the spatial distribution of surface albedo variability included in the footprint of the downward facing pyranometer did not change with increasing footprint radius. Hence, the small variations in the drone-based albedo observed from 30 m above the surface and at higher altitudes were associated to measurement errors (caused for instance by small vibrations of the drone, and small deviations from the horizontal alignment of the pyranometers when compensating for changes in wind speed) rather than to spatial variability."

18) 18. P18l412: "Hence, we can argue that it also represents the spatial albedo variability, though biased toward the albedo of the most frequent surface type that happened to occur right below the pyranometer." Can we really say here that temporal variability can be taken as a proxy for spatial variability? Albedo variability also depends on atmospheric parameters such as SZA and cloud cover, which are certainly reflected in the temporal variability within the one-month period. However, for a 10-minute flight, I would assume that these parameters have less effect.

We agree with the reviewer, our explanation was not well formulated, and indeed SZA and cloud cover played a role in the albedo variability. And we also agree that normally we cannot take temporally variability of snow/ice albedo as a proxy for spatial variability of snow/ice albedo. We therefore rewrote this part as such:

"The temporal variability of the fixed station albedo is caused by the change in solar zenith angle, the changes in cloud cover, the occurrence of precipitation and snowdrift, and, in less extent, by snow metamorphism (which was weak because the surface temperature was well below freezing for the whole period). During the studied period, the daily minimum solar zenith angle decreased from 60.5 to 56.2 deg, and cloud cover ranged from 0 to 8 oktas. The continuous snow drift, snow erosion, and formation of snow patches and dunes changed the snow thickness in the footprint area of the fixed downward looking pyranometer in a similar way as in the surrounding area. Hence, we can argue that the probability distribution of the fixed station albedo illustrates the effects of both temporal and spatial albedo variability, assuming that the snow thickness variability that occurred right below the pyranometer is a good proxy for the larger scale spatial variability."

Furthermore, we have added the mean values of the drone measurements to the sea ice albedo time series in Fig. 4.

19) 19. P18l420: "The discrepancy between the observations and the models over the thinner ice is large: Δmean = 0.2, and 0.14 for MetROMSUHel and NEMO respectively." Is the comparison meaningful, as different years with probably different conditions are taken into account?

See answer to question 16.

20) P19 Section 4.5: It is useful to show the spatial variability of satellite and model data. However, the authors could also use high-resolution satellite data to compare albedo directly with ground-based observations.

This paper is quite model-centric, and a detailed satellite to in-situ measurement comparisons are outside of the scope. Primarily, we were interested in how successfully models capture the observed conditions. We would like to leave thorough analysis of the Marsden observations to the Marsden team, who are already preparing further data releases and analysis. However, we will add satellite measurement points to the albedo time series, Fig. 4, and add to the caption:
"Sea ice albedo derived from high-resolution Landsat 9 and Sentinel 2 images is shown here and analysed further in Sect. 4.5."

And, in Sect 4.5 we make a note of it by adding:
"Overall, the albedo in the region increased between the two dates. This is also evident at the Marsden campaign site, shown in Fig. 4, where the Landsat 9 sea ice albedo is lower than the in-situ observations, while the Sentinel-2 albedo is higher. Except over the land-fast sea ice, where concentrations remain unchanged, sea ice concentrations in McMurdo Sound increased locally, which explains the observed albedo increase over the sea ice."

21) P19L439: "Landsat 9 albedo observations on the 1st of November in Fig. 7, and Sentinel 2 observations on the 14th of November, 2022 … " - Is this an example that can be used to represent the whole period?

Not the whole period. We selected cloudless high resolution satellite images from Landsat and Sentinel database. There are not many, but our aim is not to use satellite images to describe the whole period, but to describe the spatial variability of sea ice albedo and how well models reproduce it.

22) P20l444: "However, the albedo over land is about 0.06 higher in the Sentinel 2 image compared to the Landsat 9 image." – The manuscript is about sea ice. Therefore, I would limit the discussion to that.

Indeed, this was just to illustrate that the two images vary more than just over sea ice. We will omit the sentence for clarity.

23) P21l482: "This peak does not come from sea ice concentrations in the area but from the sea ice albedo parameterisation." – Can you elaborate this statement?

We have rearranged sentences and rewrote:

"RACMO has the first mode at 0.65. Figure B1 shows that sea ice concentration-independent snow and ice parameterisation over the whole region is about 0.65 for both ERA5 and RACMO. This value is consistent with Ebert and Curry (1993, Fig. 6, with Antarctic values shifted by six months), which shows sea ice albedo over Antarctica in mid-December to be around 0.6–0.7."

24) P22l489: "The spatial distribution and density distribution of the observed albedo are best reproduced by the MetROMS-UHel model." – This statement suggests that CLARA-A3 is the truth. How large is the retrieval uncertainty of the CLARA-A3 product? Perhaps it is better to say here that the MetROMS-UHel model shows the best agreement with the satellite product.

We agree that CLARA-A3 is not the truth. We have added a sentence to Sect. 2.3: "Riihelä et al. (2024) has evaluated the mean bias of the data against a selection of high-quality in situ surface albedo measurements to be generally 10%-15%."

Furthermore, we have changed the wording in the sentence in Sect 4.6:
"MetROMS-UHel model shows the best agreement with the spatial distribution and density distribution of the observed albedo." to:
"MetROMS-UHel model shows the best agreement with the spatial distribution and density distribution of the CLARA-A3 satellite albedo."

25) P24 Section Discussion: This section is more of a summary than a discussion. Apart from some text at the end that could be moved to the 'Conclusions' section, I don't see much new information here.
We will look into making the paper more compact at the expense of Sect. 5.

26) P26 Section Conclusion: It would be good to support the conclusions with some numbers to make them more quantitative.

We have added quantitative values for the points 1, 2 and 3 in the Conclusion:
  1) RACMO and ERA5 predict significantly lower albedo sea ice over the Weddell Sea during the ISPOL campaign, 0.69 and 0.63, respectively, compared to the observed 0.78 at the ISPOL radiation measurement site. This discrepancy extends across the entire Weddell Sea when compared to satellite data.

2) All models, except RACMO, predict high sea ice albedo in the range of 0.82 to 0.85, compared to the observed 0.79, due to their representation of uniform snow cover.

3) The Landsat 9 high-resolution surface albedo image of the McMurdo Sound area shows a wide distribution ranging from 0 (open ocean) to 0.9 (snow-covered sea ice), with a clear peak at 0.4. In contrast, the regional models, both the low-resolution MAR and the high-resolution HCLIM, exhibit a narrow range of albedo values and fail to capture the detailed variability observed in the satellite data.

Technical Comments

1. check format of citations for example: p2l39: "0.06.(Warren,", p2l48: "by (Debernard et al., 2017…", p3l54: "as in ERA5 Hersbach et al. (2020)", p4l107: "snow. (Hellmer et al., 2006).", p6l148: "zenith angles Traversa and Fugazza (2021)", p8l167: "scheme Lipscomb and Hunke (2004)"
2. p7 Table 1: first line "absorption/scattering" check hyphen separation, last line "othewise" typo
3. p8l194: "Melt pond properties as given by the physical level-ice scheme characterised by…" are characterised
4. P8l182: "The model runs at 0.25° resolution"  1/4° as used in Table 1
5. p9l203: "The regional atmospheric model HARMONIE Climate (HCLIM, Belušic et al. (2020)) cycle 43 using the non-hydrostatic …"  model HARMONIE Climate cycle 43 (HCLIM, Belušic et al., 2020)
6. p10 Figure 2: Please use larger fonts. Think about to show a distribution of the albedo difference in addition.
7. P11l269: "at the lateral boundaries (van Dalum et al. submitted to the Cryosphere)." Cite the discussion paper.
8. P12 Figure 3: Think about to move the legend from Fig 3c to the top of the figure.
9. P34l691: "https://doi.org/https://doi.org/10.1029/2023EA003482": remove first "https://doi.org/", there are several more references with similar issues
10. P34l700: please update reference
11. P35l738: please update reference
12. P38l841: please update reference

Thank you for pointing out the technical issues. We have addressed and corrected all of them in the revised manuscript.

In addition,
with the discussions and analysis during the peer review process, we have come across and fixed two mistakes

1)  We were using an older output version of MetROMS-UHe for Figs. 4 and 5 for the ISPOL case. The difference between the two model setups is minor with small changes in the input file of CICE, where default values were used for the newer run.

In our results, the behaviour of MetROMS-UHel at the end of the ISPOL campaign has changed, and we have made corrections in the paper. Firstly, the MetROMS-UHel has an abrupt lowering of sea ice albedo at the end of the month (around 29th of Dec), which was not present before.

Although the differences between the two model setups are minor, small discrepancies can accumulate over time, especially given that the model was run continuously from 1992, including about 16 years of simulation and spin-up by the time of the ISPOL case study.

[Figure]

Updated Fig. 4.

The value in Table 1 changes a little: 0.78 (0.08) → 0.77 (0.09)
In Fig. 5, the changes are not visible, and the snow height is not affected.

2)  The comparisons with CLARA-A3 on Fig 10-13 compared the monthly average CLARA-A3 product with model output from either 12.12.2004 (Weddell Sea) or 12.11.2022 (Ross Sea). We have updated Figs. 10-13 and B1 and B2 to compare monthly average CLARA-A3 product with model output monthly averages. The Figures changed slightly, but the conclusions stayed the same.

To address the comments here and those from Referee #2, we have added two additional figures to the manuscript compared to the previous version. As a result, the figure numbering in the revised version will differ from that referenced in this response.

We thank the reviewer again for the thoughtful and detailed comments, which have helped improve the clarity and quality of the manuscript.  We hope that the revised version meets the

reviewer's expectations, and we believe it is now significantly improved as a result of this review process.

With kind regards,
Kristiina Verro,
On the behalf of the authors.

---

## Author Comment (AC2)

**Response to the RC1 #2 comment on "How well do the regional atmospheric and oceanic models describe the Antarctic sea ice albedo?" by Verro et al.**

Thank you to anonymous Reviewer #2 for the comments, which helped identify and resolve some issues in the analysis and contributed to improving the overall quality of the article. We hereby respond to the comments point by point.

Major comments

1) While I can accept the decision to defer cloud impacts on sea ice albedo to future studies, it is on this point that the manuscript should be clearer. First, the typical difference between the clear and cloudy sky albedo should be noted, using e.g. Key (2001) as a reference. Then, if all data sources in the study are indeed clear-sky only, it should be made very clear how the models are coerced to only provide albedos consistent with clear-sky conditions.

We acknowledge that we should have taken more time to discuss the role of clouds, even if a detailed analysis is reserved for future studies.

We set out to test the modelled albedos against observations and satellite products as is. The albedo time series shown Fig. 4 shows the measured albedo during both cloudy and cloudless conditions, while the high-resolution satellite products were selected on cloudless days.

Some of the models include cloud-modified albedo (MetROMS-UHel, NEMO, MAR), while others do not (RACMO, HCLIM, ERA5). We have added modelled cloud coverage timeseries to Appendix A, and expanded Sect. 4.1:

[Figure]

"We have included total cloud coverage of the models in Fig. A3. There are considerable differences in cloud cover over the ISPOL and Marsden campaign sites between the models, which can only partly be explained by differences in resolution. The snow and ice albedo can be, on average, 4–6% higher under cloudy conditions compared to clear skies (Key et al., 2001). However, only MetROMS-UHel, MAR, and NEMO account for the effect of clouds in their sea ice albedo parameterizations, whereas HCLIM, RACMO, and ERA5 do not include this effect. "

Furthermore, we have expanded on the secondary effects on the albedo, such as cloudiness and SZA:

"Surface albedo over Polar sea ice is complex, and this study focused only on first-order effects, excluding factors such as the cloud and solar zenith angle dependence of surface albedo, which themselves can be 10% (Key 2001, Gartner and Sharp (2010), Jäkel. 2023). In study cases characterised by significant variation in surface types, such as during the spring/summer season, it is primarily uncertainties in the parameterisation of these surface types that influence the modelled surface albedo, rather than cloud effects (Jäkel. 2023). The effect of using different surface types can lead to a 20–30% difference in albedo, such as when shifting from bare ice to snow-covered ice, as seen in the case of MAR (Fig. 4b). Using a snow albedo parameterization that is not suitable for polar regions can result in differences exceeding 30%, as demonstrated by HCLIM (Fig. 2). While future research should include a more comprehensive evaluation of cloud impacts, as explored by Jäkel et al. (2023) and Foth et al. (2023), there is still room for improvement by refining the aspects of albedo discussed in this paper. "

2) Additionally, For the satellite data, the S2 and LS9 data are evidently clear-sky, though I would appreciate clearer details on the clear-sky atmospheric correction necessary to provide the surface reflectances (yes, actually these data are nadir-view directional snow reflectances – they can well be a good estimate for the view-integrated albedo, but you should be clear on the distinction). And for CLARA-A3, I think that the data there are available for various illumination conditions – which did you use here?

Thank you for your comment. We clarify below the atmospheric correction methods applied to the satellite datasets and the type of albedo used from CLARA-A3:

- Sentinel-2: We used the Level-2A product, which provides surface reflectance (bottom-of-atmosphere) already corrected for atmospheric and topographic effects using the Sen2Cor processor. This includes corrections for aerosol optical thickness (AOT), water vapor, and terrain effects, based on Look-Up Tables generated via libRadtran and adapted from the ATCOR software (Richter et al., 2006). The baseline aerosol model is rural/continental, with atmospheric profiles selected according to scene location and climatology (Main-Knorn et al., 2017).

- Landsat 9: Surface reflectance was derived following the method described in Traversa et al. (2021), using the 6S radiative transfer model (Vermote et al., 1999). Inputs included satellite geometry, date/time/location, a subarctic winter atmospheric model, a continental aerosol model, and visibility or AOT at 550 nm.

- CLARA-A3: As stated in the manuscript, we used the blue-sky albedo product, which represents a weighted combination of black-sky (direct) and white-sky (diffuse) albedo, based on the estimated clear-sky fraction. This choice provides a more realistic approximation of surface albedo under typical illumination conditions.

We acknowledge the importance of distinguishing between nadir-view directional reflectance (as provided by S2 and L9) and hemispherical albedo (as in CLARA-A3). While directional reflectance can serve as a proxy for albedo under certain conditions (e.g., over snow), we have taken care to interpret and compare these datasets accordingly.

3) 4.4 – it's not clear how many drone flights contributed to the histograms in Fig 6? If from multiple days, what was the day-to-day albedo variability in the drone data? Were the flights made always over the same survey grid? Was the weather clear or cloudy – that would also change the observed snow albedo.

The histograms were done during a single flight over each surface type. The measurements were taken between 30 and 50 m over thick ice (~2.1 m, CS) and between 30 m to 70 m altitude over thin ice (~1.2 m, NIS), which are shown on the map on Fig. 1.

We have added more explanations on the drone measured albedo:.

"Figure 6a-b shows the probability distributions of the albedo measured from a drone flying a single vertical profile at an altitude between 30 and 50 m over thick ice and another profile between 30 m to 70 m altitude over thin ice with patchy snow cover. The flight lasted ~10 min, without significant changes in solar zenith angle and cloud conditions."

Therefore, the distribution of the drone-based albedo data did not include day-to-day variability, and cloud conditions were unchanged during the flights. Furthermore, we have added the mean values of the drone measurements to the sea ice albedo time series of Fig. 4.

4) 4.5. – you can calculate the mean SZA over the study areas as you know the S2 and LS9 overpass times; a first-order estimate for the albedo effect assuming non-melting snow would then easily be available from a lightweight albedo parameterization such as that of Gardner and Sharp (2010), please provide the assessment in the text as a yardstick for the reader.

For L9: SZA = 70.8°, overpass time: 20:21:00.7699389Z, and for S2: SZA = 67.3°, overpass time: 20:05:29.024Z.  As suggested, we have estimated the albedo effect of SZA from Gadner and Sharp (2010) and have provided the yardstick albedo effects in the text:

"We can estimate the effect of solar zenith angle on snow and ice albedo between the two dates using parameterisation from Gartner and Sharp (2010). Assuming pure, dry fresh snow with an albedo of 0.9 and a solar zenith angle of $0^0$, the albedo reduction due to the solar zenith angle at the Landsat 9 overpass (SZA = $70.8^0$) is 0.038. For the Sentinel 2 overpass (SZA = $67.3^0$), the corresponding albedo effect is 0.036. For ice and snow with lower albedo values, the effect is more pronounced -- for example, sea ice with an albedo of 0.5 experiences albedo effects of 0.1, for both SZA = $70.8^0$ and $67.3^0$. However, the difference in albedo between the two observation dates remains small, on the order of 0.01 or less."

5) Figs 10-12: It is my understanding that CLARA-A3 (not 3A as in some legends) albedos are either 5-day or monthly means, yet here the text refers to CLARA-A3 products from a specific day. Did you recompute your own daily version based on provided raw data?

Thank you for your observation. You are correct that the CLARA-A3 SAL dataset provides surface albedo products as either monthly means or 5-day means, depending on the product type. In our case, we used the polar monthly mean product, and the dates mentioned in the text refer to the central date of the monthly averaging period, as defined in

the file naming convention. We did not recompute any daily albedo values from raw CLARA-A3 data.

This comment drew attention to a mistake on our part: the monthly mean satellite product was indeed compared to the daily values of the date in the CLARA-A3. We have changed the Figures 10-13.
The spatial patterns of sea ice albedo changed little, especially over the Ross Sea during November 2022, but it does not change any of the conclusions drawn.

We added this information to the text in line 204: "Furthermore, the CLARA-A3 blue-sky albedo monthly product (dates mentioned in the text refer to the central date of the monthly averaging period, Karlsson et al., 2023), which has a coarser resolution of 25 km, …"

We have made a further clarification in the caption in the Figures 10, 11: "Monthly mean ERA5 (a) reanalysis and CLARA-A3 (b) satellite albedo products from over the Weddell Sea domain as reference for model validation, and corresponding monthly mean albedo maps from MetROMS-UHel (c), NEMO (d), HCLIM (e), MAR (f) and RACMO (g) models."

Old Fig. 10:

[Figure]

Updated Fig. 10:

[Figure]

Old Fig. 11:

[Figure]

Updated Fig. 11:

[Figure]

We have fixed the "3A" typos in legends and in text.

324: 20 cm of snow is considered thin? From the optical (albedo) viewpoint, 10 cm is typically enough to effectively make the snowpack optically semi-infinite. The text does refer to this effect (for both ice and snow), but it would be nice to have quantified estimates here for typical depths required – and for the authors to consider if any of the results are affected. Also, while melt ponds are rarely encountered over Antarctic sea ice, it would be nice from the completeness viewpoint to recall that melt pond albedo is also not uniform, but depends on the depth of the pond and the properties of the underlying ice. Several appropriate references exist highlighting this effect.

The snow thickness of 0.2 m is considered thin mainly in comparison with the snow thickness observed over the Weddell Sea. But the reviewer is right, 0.2 m is sufficient to make the snow optically semi-infinite. We therefore corrected one sentence, and added one additional sentence to clarify this point:

"The snow on top of the 1 to 4 m thick land-fast ice in the McMurdo Sound had reportedly relatively thin (~0.21 m, CS), or patches of thin (~0.02 m, NIS) snow on top. Snow thickness

variability in the observed range between 0 and 0.2-0.3 m has a big impact on the albedo. For densely packed, fine-grained snow, the snow layer becomes semi-infinite—meaning that further increases in depth no longer affect snow reflectance—at a depth of approximately 0.10 m"

Minor comments:

131: A +/-1% measurement uncertainty sounds very high for field conditions. Is this a manufacturer estimate?

In fact, this albedo uncertainty is the lowest that can be possibly reached with the most accurate and well calibrated pyranometers, such as the Kipp&Zonen CMP22 used for this campaign. It corresponds to the sum of the different error sources as estimated by the manufacturer.

fig 3: legend gets lost in the subplot c, please consider moving it outside of plot area.

We have done so.

In addition,
the work done during the peer review process revealed a mistake in the analysis. We were using an older output version of MetROMS-UHe for Figs. 4 and 5 for the ISPOL case. The difference between the two model setups is minor with small changes in the input file of CICE, where default values were used for the newer run.

In our results, the behaviour of MetROMS-UHel at the end of the ISPOL campaign has changed, and we have made corrections in the paper. Firstly, the MetROMS-UHel has an abrupt lowering of sea ice albedo at the end of the month (around 29th of Dec), which was not present before.
Although the differences between the two model setups are minor, small discrepancies can accumulate over time, especially given that the model was run continuously from 1992, including about 16 years of simulation and spin-up by the time of the ISPOL case study.

[Figure]

Updated Fig. 4.

The value in Table 1 changes a little: 0.78 (0.08) → 0.77 (0.09)
In Fig. 5, the changes are not visible, and the snow height is not affected.

To address the comments here and those from Referee #1, we have added two additional figures to the manuscript compared to the previous version. As a result, the figure numbering in the revised version will differ from that referenced in this response.

We thank the reviewer again for the thoughtful comments, which have helped improve the clarity and quality of the manuscript. We hope that the revised version meets the reviewer's expectations, and we believe it is now significantly improved as a result of this review process.

With kind regards,
Kristiina Verro,
On the behalf of the authors.